# CK1/Doubletime activity delays transcription activation in the circadian clock

Deniz Top[1]*, Jenna L O'Neil[1], Gregory E Merz[2], Kritika Dusad[2], Brian R Crane[2], Michael W Young[1]

[1]Laboratory of Genetics, The Rockefeller University, New York, United States; [2]Department of Chemistry and Chemical Biology, Cornell University, New York, United States

**Abstract** In the *Drosophila* circadian clock, Period (PER) and Timeless (TIM) proteins inhibit Clock-mediated transcription of *per* and *tim* genes until PER is degraded by Doubletime/CK1 (DBT)-mediated phosphorylation, establishing a negative feedback loop. Multiple regulatory delays within this feedback loop ensure ~24 hr periodicity. Of these delays, the mechanisms that regulate delayed PER degradation (and Clock reactivation) remain unclear. Here we show that phosphorylation of certain DBT target sites within a central region of PER affect PER inhibition of Clock and the stability of the PER/TIM complex. Our results indicate that phosphorylation of PER residue S589 stabilizes and activates PER inhibitory function in the presence of TIM, but promotes PER degradation in its absence. The role of DBT in regulating PER activity, stabilization and degradation ensures that these events are chronologically and biochemically linked, and contributes to the timing of an essential delay that influences the period of the circadian clock.
DOI: https://doi.org/10.7554/eLife.32679.001

*For correspondence: dtop@rockefeller.edu

**Competing interests:** The authors declare that no competing interests exist.

## Introduction

The proteins that comprise circadian clocks are largely conserved across the animal kingdom (*Crane and Young, 2014*; *Young and Kay, 2001*; *Zheng and Sehgal, 2012*). These clocks, which promote daily rhythms in behavior and physiology, involve cell-autonomous, transcription-translation feedback loops that oscillate with ~24 hr periodicity. In *Drosophila*, the activator complex consists of dClock/Cycle (dCLK/CYC), which initiates transcription of *period* (*per*) and *timeless* (*tim*). PER and TIM proteins assemble to form an inhibitory complex in the cytoplasm that, after a delay, enters the nucleus to inhibit dCLK/CYC activity. After a second delay, the inhibitory complex is degraded, freeing the activator complex to reinitiate transcription, closing the feedback loop. Although the delay in transcriptional activation is critical for maintaining ~24 hr rhythmicity, its regulatory mechanisms are not well understood.

A Casein Kinase I/Doubletime (DBT) phosphorylation cascade on PER regulates PER protein stability. Current models of this phosphorylation cascade infer that each DBT phosphorylation event gates later phosphorylation events on PER, thus delaying protein degradation and transcriptional activation. The cascade begins with the PER Short Downstream region (PER-SD, amino acids 604–629), which promotes S47 phosphorylation (*Garbe et al., 2013*). Phosphorylation of S47, is a necessary modification to trigger ubiquitination and subsequent degradation of PER by the E3-ligase SLIMB (*Chiu et al., 2008*; *Garbe et al., 2013*; *Kivimäe et al., 2008*). PER-SD also blocks NEMO kinase from phosphorylating S596. By phosphorylating S596, NEMO promotes S589 phosphorylation by DBT, which blocks S47 phosphorylation (*Chiu et al., 2011*; *Garbe et al., 2013*). In other words, PER-SD phosphorylation removes the S589-mediated negative regulation of S47

**eLife digest** Many behaviors, such as when we fall asleep or wake up, follow the rhythm of day and night. This is regulated in part by our 'circadian clock', which controls biological processes through the timed activation of hundreds of genes over the 24-hour day.

In fruit flies, the proteins that form the core of the circadian clock activate and repress each other in such a way that their expression oscillates over a 24-hour cycle. During the late afternoon and early evening, the Clock protein initiates the production of proteins Period and Timeless: these two molecules then accumulate in the cell, and after binding to each other, they are transported into the nucleus. During the late night and early morning, this Period/Timeless complex inhibits the activity of Clock. After a delay, Period and Timeless are degraded. This allows Clock to be reactivated, restarting the cycle for the next day.

Period is critical to help maintain the 24-hour oscillation shown by these proteins. A protein called Doubletime is responsible for making a number of chemical modifications on Period. It is unclear how these changes interact with each other, and how they influence the stability and function of Period when it is associated with Timeless.

Here, Top et al. generate mutations in the fruit fly gene *period* to study these processes, and develop a new biomolecular technique to monitor the stability and activity of Period protein in insect cells grown in the laboratory.

The experiments reveal new roles for the chemical changes made by Doubletime to Period. First, after Period associates with Timeless, Doubletime triggers certain modifications that lead to Period being able to inactivate Clock. Second, Doubletime makes another change in a nearby region of Period that results in the Period/Timeless complex being stabilized. Both sets of modifications help the complex to stay active and keep inhibiting Clock for long enough such that a 24-hour rhythm can be maintained. Finally, when Timeless is degraded, Period is released from the complex. At this time, the modifications made by Doubletime promote the degradation of Period, resetting the clock.

Fruit flies with mutations that block this mechanism perceive the day as shorter. This shows that the smallest change to clock genes can disorganize behavior. Indeed in humans, health problems such as sleep or mental health disorders are associated with irregular circadian clocks. Understanding the biochemical mechanisms that keep the body clocks ticking could help to find new therapeutic targets for these conditions.

DOI: https://doi.org/10.7554/eLife.32679.002

phosphorylation while independently promoting S47 phosphorylation (*Garbe et al., 2013*). The purpose of the second regulatory route involving negative regulation of degradation by S589 is unclear.

Mutation of the S589 site in flies leads to a short behavioral period (*Baylies et al., 1992*; *Konopka and Benzer, 1971*; *Rutila et al., 1992*). Given that S589 is a DBT phosphorylation site (*Chiu et al., 2008*; *Kivimäe et al., 2008*), it is surprising that mutations that both block (Gly) and substitute for phosphorylation (Asp) lead to a short behavioral rhythm (*Rutila et al., 1992*). Although this may suggest that the aspartate residue substitution does not serve as an effective phosphomimetic, the fact that the behavioral period of the mimetic is longer than a mutant that prevents phosphorylation (Gly) suggests a more complex regulatory mechanism.

TIM binding to PER adds another layer of complexity to the model by protecting PER from hyperphosphorylation and degradation (*Kloss et al., 2001*). This suggests that the DBT phosphorylation cascade on PER is influenced by TIM. The sites that comprise the cascade were identified using either purified protein (*Kivimäe et al., 2008*) or PER protein expressed alone, in cultured Drosophila embryonic (S2) cells (*Chiu et al., 2008*), concealing the identity of the sites that would have been protected by TIM, or overlooking sites that would have been phosphorylated in the presence of TIM. Thus, how the DBT phosphorylation cascade operates in the context of a complex is largely undetermined.

It is generally assumed that inhibition of dCLK is gated by nuclear accumulation of PER, and release of dCLK inhibition occurs passively, through PER degradation. PER phosphorylation, PER degradation and PER inhibitory activity coincide closely in time and space (*Abruzzi et al., 2011*;

*Edery et al., 1994*), which has led to a model in which PER stability and inhibitory activity are regulated by the same regulatory mechanism (*Kivimäe et al., 2008*). An alternative possibility is that the different functions of PER (transcription inhibition and degradation), are independently regulated by different programs of phosphorylation that occur concurrently or nearly so. This possibility has been difficult to test, due to the challenges of monitoring PER phosphorylation, transcriptional activity and protein stability simultaneously.

In this study, we propose a new model for the regulation of PER by DBT, which includes separable features affecting PER stability and activity. Our findings suggest that the DBT/PER/TIM complex is interdependent in the initial stages of dCLK/CYC inhibition. We show that the initial sites of DBT phosphorylation stabilize the PER/TIM inhibitor complex and activate transcriptional inhibition. The classic per short site S589 serves two functions: first to stabilize the activated PER inhibitor in the presence of TIM and then to mediate its degradation after TIM is degraded. We propose a dual role for DBT in regulating the circadian clock. (1) In the nucleus DBT activates function of the PER/TIM complex as a transcriptional inhibitor and stabilizes it, (2) subsequently promoting PER degradation by midday. The dual function of DBT links PER/TIM inhibitory activity and stability, biochemically and chronologically, and underlies a key delay in the circadian clock that helps establish 24 hr behavioral rhythmicity.

## Results

### PER S589 substitutions are associated with short-period rhythmic behavior

PER S589 is involved in a phosphorylation cascade that regulates PER ubiquitination and degradation (*Chiu et al., 2008*; *2011*; *Garbe et al., 2013*; *Kivimäe et al., 2008*). Paradoxically, residue substitutions of S589 that either block or mimic phosphorylation both yield the same phenotype: short-period rhythmic behavior in transgenic flies (*Baylies et al., 1992*; *Rutila et al., 1992*). To confirm this, we generated transgenic flies in which we rescued the $per^0$ mutation with $per$ variants containing substitutions of the DBT-targeted $per^S$ site S589, using targeted genomic integration to avoid positional effects on protein expression. As previously reported (*Baylies et al., 1992*; *Rutila et al., 1992*), a phosphonull alanine substitution (S589A) caused an advance in evening anticipation but not morning anticipation relative to controls in a 12:12 light/dark (LD) light regiment (*Figure 1A*). In constant darkness (DD), S589A mutants exhibited a ~ 6 hr shorter period in activity rhythm relative to controls (*Figure 1B*, *Supplementary file 1*; ~19.5 hr S589A vs ~25.5 hr wild type).

To test the effects of constitutive phosphorylation at residue S589 on period length, we generated a PER variant with the phosphomimetic aspartate substitution at site S589 (S589D). Similar to phosphonull S589A mutants, we found that phosphomimetic S589D mutants exhibit a modest advance in evening anticipation and no change in morning anticipation relative to controls (*Figure 1A*). Consistent with this result, the activity period of S589D in DD was also shortened, leading to a ~ 2 hr shorter subjective day (*Figure 1B*, *Supplementary file 1*; 23.5 h S589D vs ~25.5 hr wild type). Thus, we confirmed that mutations that mimic or block phosphorylation both have the effect of shortening behavioral period by advancing evening anticipation. Paradoxically, these results suggest that both phosphorylation and loss of phosphorylation at S589 normally contribute to the post-transcriptional delays that regulate circadian period length.

### S589 mutants block changes to rhythmic behavior caused by DBT overexpression

It was not clear whether these effects on period length by this residue is solely due to phosphorylation by DBT kinase or whether there may be a second kinase involved. We can distinguish between these two possibilities by testing genetic interactions between specific phosphomutations and kinase overexpression or kinase mutant expression. For example, mutation of specific target residues of the TIM protein blocks the effect of CK2 kinase overexpression on rhythmic behavior (*Top et al., 2016*). DBT overexpression in transgenic flies expressing wild type PER caused a modest increase in period length of ~1 hr, as expected (*Muskus et al., 2007*; *Venkatesan et al., 2015*). We found that both S589 mutations blocked this effect of DBT overexpression on period length (*Figure 1C*). Since exogenous DBT expression has a phenotypic effect on period length in wild type flies, endogenous DBT

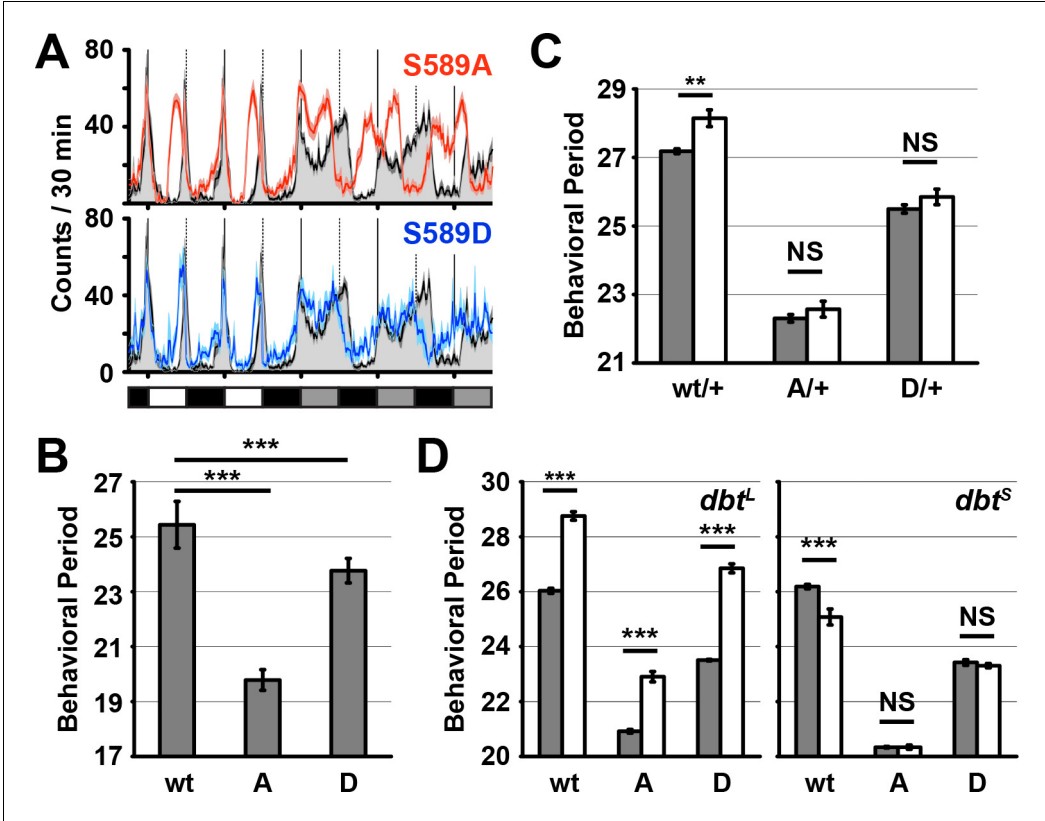

**Figure 1.** Residue substitutions of S589 differently affect behavioral rhythmicity. (**A**) Transgenic wild-type *per* (black line), S589A *per* (red line) or S589D *per* (blue line) were expressed in a *per⁰* genetic background and monitored for behavior in a 12:12 light dark cycle followed by constant darkness and activity recorded and plotted. The solid vertical lines demark days, while the dotted vertical lines represent the light-to-dark transition point. White, black and grey boxes represent lights on, lights off and subjective day during lights off, respectively. N = 27–28. Each data point is the mean ± SEM. Thickness of the curve reflects SEM. (**B**) Flies described in panel A quantified for behavioral periodicity in constant darkness. wt: wild type, A: S589A, D: S589D. N = 25–27. Bars represent means ± SD. \*\*\*p<0.005. Significance was measured by an unpaired, two-tailed t-test. See *Supplementary file 1* for values. (**C**) The behavioral period of heterozygous flies carrying the transgenic *per* variant (as indicated) in the absence of DBT overexpression (grey bars, *per⁰*; attP40{*per*} / +; UAS-*dbt*) or with DBT overexpression in clock neurons (white bars, *per⁰*; attP40{*per*}/*tim*-UAS-Gal4; UAS-*dbt*) were quantified and plotted. N = 19. Bars represent means ± SD. \*\*p<0.01. NS, not statistically significant. Significance was measured by t-test. See *Supplementary file 1* for values. (**D**) The behavioral period of *per* transgenic flies in a *dbt^L* or *dbt^S* genetic background (white bars) or wild type *dbt* background (grey bars) were quantified and plotted. N = 8–24. Bars represent means ± SD. \*\*\*p<0.005. NS, not statistically significant. Significance was measured by t-test. See *Supplementary file 1* for values.

DOI: https://doi.org/10.7554/eLife.32679.003

The following figure supplement is available for figure 1:

**Figure supplement 1.** Residue substitutions of S589 do not genetically interact with *O*-Glc-NAc Transferase (OGT).

DOI: https://doi.org/10.7554/eLife.32679.004

---

levels are rate limiting for mechanisms affecting behavior. The ability of S589 to block the effect of DBT overexpression on behavior suggests that its modification occurs before other functions of DBT that regulate behavior. The ability of S589 to curb the effect of DBT overexpression also provides evidence that S589 and DBT genetically interact.

The activity of *O*-GlcNAc transferase (OGT) has been shown to influence behavioral rhythmicity in mammals and flies, either by competing with CKIδ for modification of mPER2 (orthologues of DBT and PER, respectively) or blocking BMAL1/CLOCK (orthologues of CYC/dCLK in flies) ubiquitination

in mice, or prolonging PER stability in flies (*Kaasik et al., 2013*; *Kim et al., 2012*; *Li et al., 2013*). Since we were able to successfully block the phenotypic effect of DBT overexpression in the S589A and S589D fly mutant backgrounds, we decided to test whether the S589 residue genetically interacts with OGT. We found that OGT overexpression in wild type flies lengthened rhythmic behavior of wild type flies, as expected (*Kaasik et al., 2013*; *Kim et al., 2012*). The same experiment conducted in the S589A and S589D mutant backgrounds similarly lengthened rhythmic behavior of the mutant flies (*Figure 1—figure supplement 1*). We therefore conclude that S589 does not genetically interact with OGT.

Mutant forms of *dbt* that reduce DBT kinase activity produce long ($dbt^L$) and short ($dbt^S$) behavioral rhythms (*Kivimäe et al., 2008*; *Preuss et al., 2004*; *Price et al., 1998*), suggesting that DBT may play two roles in regulating PER. Indeed, in cultured cells $dbt^L$ is defective in regulating PER stability, whereas $dbt^S$ is not associated with regulating PER stability (*Syed et al., 2011*). To test whether S589 mutants genetically interact with DBT function affecting PER stability, we assayed the effects of combining S589 mutations with either $dbt^L$ and $dbt^S$. We found that expression of either phosphonull (S589A) or phosphomimetic (S589D) *per* variants in a $dbt^L$ genetic background leads to longer behavioral rhythms (*Figure 1D*). In contrast, expression of wild type *per* in a $dbt^S$ genetic background leads to a shorter behavioral rhythm of wt flies, but not of the S589 mutants. Thus, mutations of S589 block the period changes caused by $dbt^S$ but not $dbt^L$, suggesting that the S589 mutations and $dbt^S$ lie within the same regulatory pathway.

## PER S589 is not involved in regulating PER/TIM nuclear accumulation

We set out to identify the mechanism by which PER phosphomutants interact with $dbt^S$ to delay the circadian clock. Delays in the circadian clock are regulated by post-translational modifications that govern PER nuclear entry, PER inhibitory activity, and PER/TIM degradation. $dbt^S$, which has reduced kinase activity, delays PER nuclear accumulation in vivo (*Bao et al., 2001*; *Kivimäe et al., 2008*; *Rothenfluh et al., 2000*). Although the original PER$^S$ mutant protein (S589N) exhibits kinetics of nuclear entry similar to wild type (*Curtin et al., 1995*), because S589 phosphomutants and $dbt^S$ genetically interact, we first tested whether S589 phosphomutations alter the kinetics of PER/TIM nuclear accumulation. We quantified PER protein expression using immunofluorescence microscopy analysis across the circadian cycle, with focus on the master pacemaker neurons (the ventral lateral neurons; LNvs) that drive behavioral rhythms in constant darkness (*Stoleru et al., 2005*) (*Figure 2A*). Similar to the original PER$^S$ mutant, we found that S589 mutants did not alter the kinetics of nuclear accumulation of PER or TIM in the small LNvs (*Figure 2B*) or the large LNvs (*Figure 2—figure supplement 1*) relative to controls. Thus, residue substitutions of S589 do not shorten period length by changing the rate of nuclear accumulation of the PER/TIM inhibitory complex. This result further suggests that S589 phosphorylation regulates PER function after nuclear entry.

## Phosphorylation of PER S589 stabilizes the PER/TIM complex in the small LNvs

We next set out to determine the role of S589 in PER and TIM protein stability after nuclear entry. Using in vivo fluorescence microscopy, we examined PER/TIM protein stability in the master pacemaker small LNvs, from ZT20 through CT08, when the complex is predominantly nuclear in wild type flies (*Figure 2C*). The phosphonull S589A substitution does not affect PER stability between ZT20 and CT02, but destabilizes PER after CT02, leading to a gradual loss of PER protein relative to controls (*Figure 2D*). TIM is similarly destabilized in the S589A genetic background compared to wild type (*Figure 2E*). In contrast, the phosphomimetic S589D mutant behaves like wild-type PER until TIM is degraded at CT06, at which time the PER S589D protein is rapidly degraded compared to wild type (*Figure 2D and E*). The changes in the stability of the PER variants correlate well with the differences in rhythmic behavior exhibited by the relevant fly mutants (*Figure 1B*). These data suggest that S589 phosphorylation is required to stabilize the PER/TIM inhibitory complex after CT02 in the small LNvs. However, upon TIM degradation, the behavior of S589D indicates that phosphorylation at S589 now promotes PER degradation, ending the transcriptional inhibitory phase of the circadian clock. That is, phosphorylation at S589 both stabilizes and destabilizes PER at different points in the circadian cycle, depending on the presence or absence of TIM protein. Thus, S589 phosphorylation has a stabilizing effect when TIM is present.

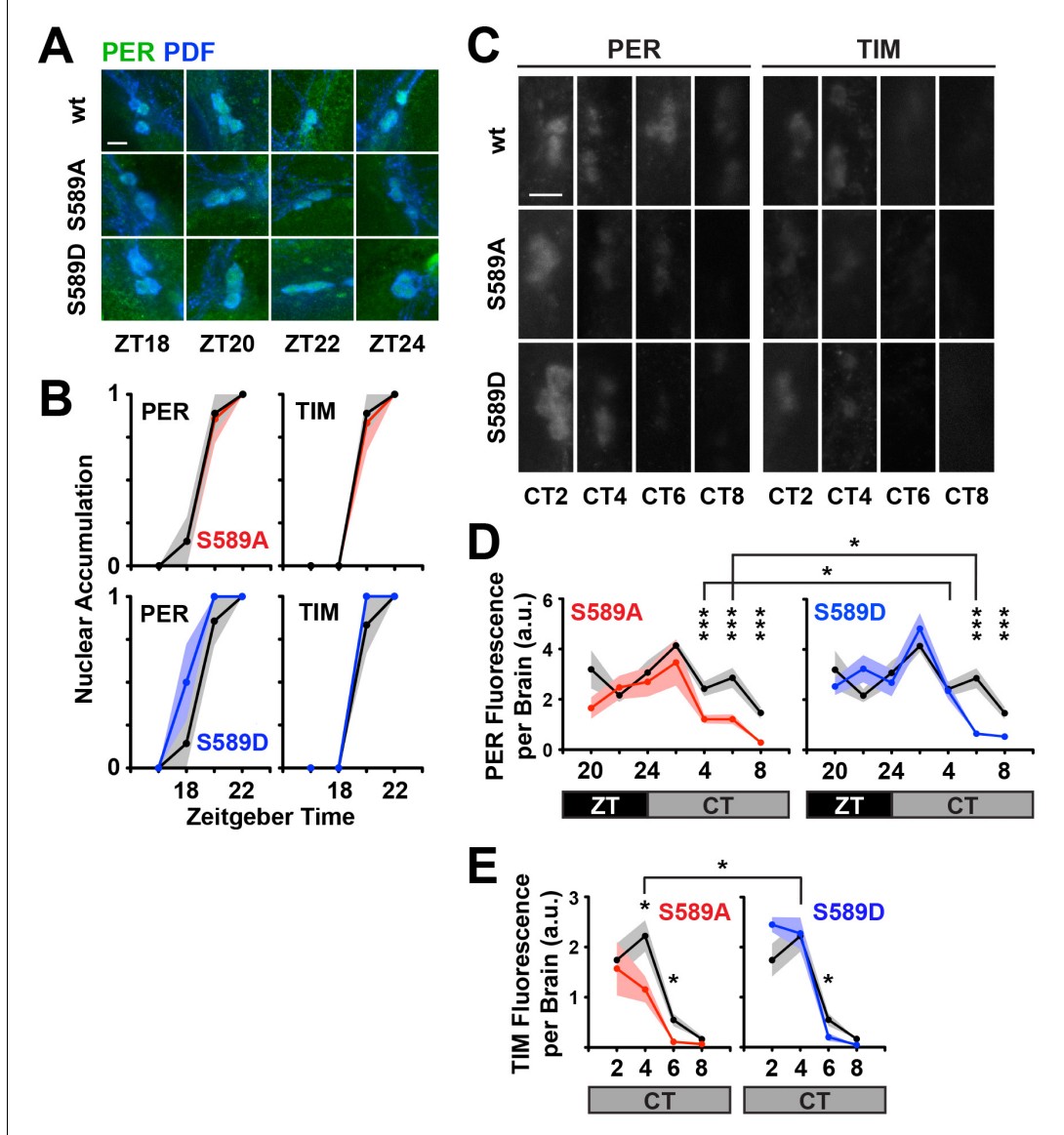

**Figure 2.** Residue substitutions of S589 do not affect nuclear entry, advance PER degradation in small ventral lateral neurons in vivo. (**A**) Fly brains were collected at the indicated zeitgeber time points (ZT) and fixed. Representative images of s-LNvs stained for PER (green) and PDF (blue) are shown. The scale bar represents 5 μm. (**B**) Stained fly brains were visually scored to determine PER/TIM nuclear accumulation. PER or TIM staining that is predominantly nuclear or cytoplasmic in the small LNvs were valued 1 or 0, respectively. Six to ten fly brains were averaged for each mutant (S589A, red line; S589D, blue line) at the indicated time points, and compared to wild type control (black line). Each data point represents the mean, ±SEM. Thickness of the curve reflects SEM. (**C**) Fly heads were collected at the indicated circadian time points (CT; constant darkness). Representative images of s-LNvs stained for PER and TIM are shown. PDF stain is omitted for clarity of PER/TIM signal intensity. The scale bar represents 5 μm. (**D**) Stained fly brains were quantified for PER expression in small LNvs. Fluorescence signal from 6 to 10 fly brains were imaged and quantified (arbitrary units, a.u.) for each time point, normalized to an internal standard and the two hemispheres averaged to represent the brain. Mutant PER expression (S589A, red line; S589D, blue line) and wild type PER (black line) was plotted and compared. Times are measured in ZT (black box) or in CT, subjective day (grey box). Each data point represents the mean, ±SEM. Thickness of the curve reflects SEM. Black line compares the S589A and S589D mutants. *p<0.05. ***p<0.005. Data points with no asterisks are not statistically significant. Significance was measured by t-test. (**E**) Stained fly brains were quantified for TIM fluorescence signal as described in panel D. Each data point represents the mean, ±SEM, plotted across CT. Thickness of the curve reflects SEM. Black line compares the S589A and S589D mutants. *p<0.05. Significance was measured by t-test.

DOI: https://doi.org/10.7554/eLife.32679.005

The following figure supplement is available for figure 2:

**Figure supplement 1.** Residue substitutions of S589 do not affect nuclear accumulation in the large ventral lateral neurons in vivo.

DOI: https://doi.org/10.7554/eLife.32679.006

# S589 mutants alter PER stability in a minimal inhibitory complex with TIM and DBT

DBT activity regulates PER stability by targeting S589 and other sites on PER, directing DBT to block or promote S47 phosphorylation mediated PER degradation (*Chiu et al., 2008*; *Garbe et al., 2013*; *Kivimäe et al., 2008*). We wanted to determine if mutating S589 affects the stability of the PER/TIM complex coupled to DBT activity. PER/TIM complex stability can be directly investigated using S2 cells, which offer a minimal system uncomplicated by an endogenous clock that could alter protein expression through transcriptional regulation. Analysis of the S589N variant of PER (PER$^S$) in a similar study that included TIM (but not DBT) in cultured cells suggests that it is equivalent in stability to wild type PER (*Li and Rosbash, 2013*). However the side chain of asparagine has a dipole that has the potential to act as a phosphomimetic while simultaneously blocking phosphorylation by DBT. Therefore to further characterize the effect of the S589 substitutions on the stability of the nuclear PER/TIM complex, we performed pulse-chase co-expression experiments in which *per* or *per* variants were expressed by a heat-shock promoter and *tim* and *dbt* were co-expressed using a constitutively active actin promoter. In this experimental design, PER and associated proteins are predominantly nuclear because driving *tim* expression by an actin promoter bypasses the delay in nuclear accumulation (*Figure 3—figure supplement 1A*) observed when both *tim* and *per* are driven by heat-shock promoter (*Meyer et al., 2006*; *Saez et al., 2011*; *Top et al., 2016*). S2 cells were lysed at 1 hr time resolution and analyzed by quantitative western blot (*Figure 3*).

The analysis was plotted as ratios to normalize against different factors that affect protein stability. Because pulse-chase experiments involving cycloheximide display gradual loss of all proteins, as a control we normalized PER and TIM protein decay to DBT, whose stability is not under circadian clock regulation (*Figure 3—figure supplement 1B–C*). By using DBT as a proxy for background degradation, we control for the effect of cycloheximide on the kinetics of general protein degradation. The resulting plots reveal a stable S589D variant compared to wild type, and an unstable S589A variant in the presence of wild type DBT (*Figure 3—figure supplement 1C*). However, PER interactions with a DBT variant lacking kinase activity (DBT-K38R) stabilizes PER (*Muskus et al., 2007*), suggesting that competitive binding of PER with DBT-K38R versus endogenous DBT should have a stabilizing effect on PER. To isolate DBT activity as the regulating factor in PER stability, we normalized PER and TIM levels co-expressed with DBT to PER and TIM levels co-expressed with DBT-K38R (*Figure 3*). Thus, our approach normalizes PER and TIM degradation to both background kinetics of protein degradation, accounts for loss of DBT and for the stabilizing effect of DBT-PER protein interactions, effectively isolating the phosphorylation effect of DBT on PER (and TIM) stability.

Using this assay, we found that the phosphonull S589A mutation destabilizes PER relative to wild-type PER in the presence of DBT (*Figure 3A*). This result suggests that phosphorylation of S589 on PER normally protects PER from DBT-mediated degradation in a complex with TIM. This result also suggests that S589 phosphorylation is not required to complete the PER phosphorylation cascade to promote PER degradation. Consistent with this finding, PER protein containing the phosphomimetic S589D mutation exhibits stability similar to wild type in the presence of DBT activity (*Figure 3A*). The S589D result also suggests that DBT activity inhibits PER degradation at this stage. TIM stability in this assay follows the same pattern as the PER variants; destabilized by the PER phosphonull variant (S589A) but not the PER phosphomimetic (S589D), compared to controls (*Figure 3B*). This assay repeated in the absence of TIM reveals rapid PER degradation that is not quantifiable over time (*Figure 3—figure supplement 1D*). These data therefore suggest that S589 phosphorylation stabilizes PER in the presence of TIM from DBT-mediated degradation, either by blocking phosphorylation of other PER destabilizing sites, or in spite of phosphorylation of these other sites. In contrast, blocking S589 phosphorylation redirects DBT to promote the degradation of the PER/TIM complex, likely by phosphorylating other PER sites. Therefore S589 acts to direct DBT to degrade PER or to stabilize PER within the PER/TIM complex.

DBT-mediated degradation of PER occurs in both the nucleus and cytoplasm (*Price et al., 1998*; *Venkatesan et al., 2015*). The experimental system above investigates PER stability mainly in the nucleus. To determine if S589 phosphorylation affects PER stability in the cytoplasm, we repeated the pulse-chase experiments in conditions that restricted PER to the cytoplasm using a TIM mutant lacking its nuclear localization signal (TIM-ΔNLS). This TIM variant blocks nuclear accumulation of both TIM and PER in S2 cells in the time scales used in our assay (*Saez et al., 2011*). When PER and

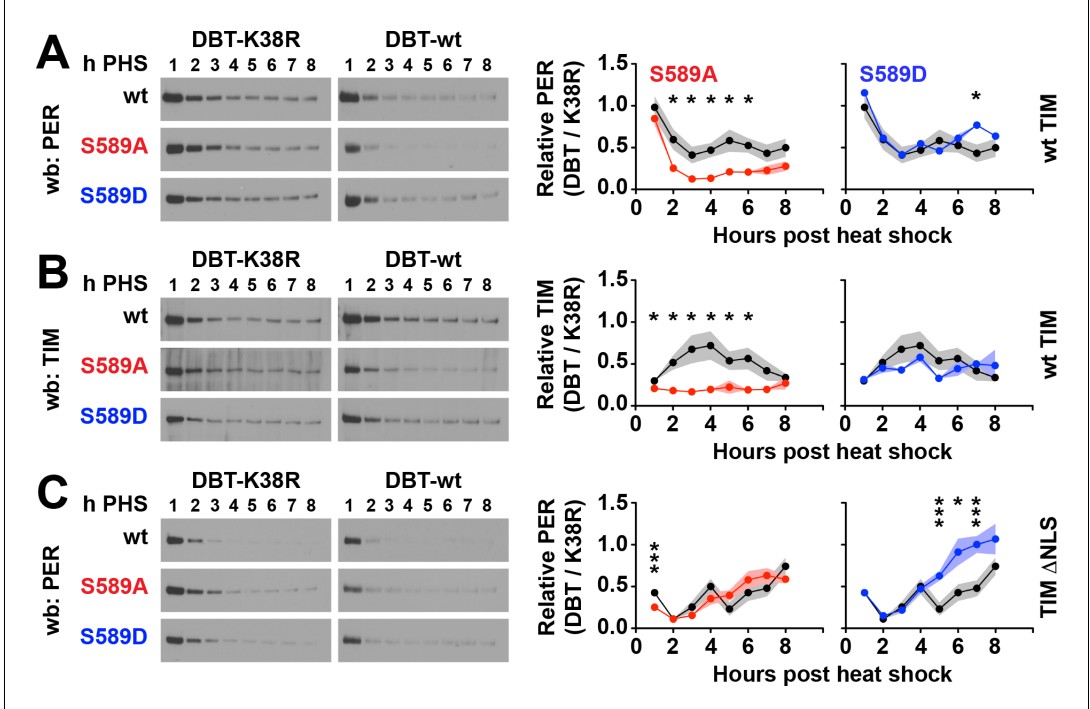

**Figure 3.** Residue substitutions of S589 determine PER/TIM stability in cultured cells. S2 cells expressing *per* driven by a heat-shock promoter, and *dbt* and *tim* driven by actin promoter were collected each hour post heat shock (h PHS) in a pulse-chase experiment using cycloheximide. (**A**) Wild type, S589A or S589D *per* variants were co-expressed in S2 cells with wild type (wt) TIM and either wild type DBT (DBT-wt) or mutant DBT lacking ATPase activity (DBT-K38R). Representative western blots (wb) are shown. PER signal was quantified and normalized to either DBT-wt or DBT-K38R signal, and the ratios plotted. S589A (red) and S589D (blue) was compared to wild type PER (black). N = 4. Each data point represents the mean, ±SEM. Thickness of the curve reflects SEM. *p<0.05. ***p<0.005. Data points without asterisks are not statistically significant. Significance was measured by t-test. (**B**) The same analysis conducted in panel A was repeated for TIM co-expressed with PER, S589A or S589D, and DBT-wt or DBT-K38R. (**C**) The same analysis conducted in panel A was repeated for PER, with a mutant form of TIM lacking a nuclear localization signal (TIM ΔNLS).
DOI: https://doi.org/10.7554/eLife.32679.007

The following figure supplement is available for figure 3:

**Figure supplement 1.** PER/TIM nuclear accumulation in S2 cells is promoter-dependent, PER stability is plotted as a PER/DBT ratio.
DOI: https://doi.org/10.7554/eLife.32679.008

TIM are restricted to the cytoplasm, wild type PER and PER S589A are similar in their stability (*Figure 3C*). Thus, lack of phosphorylation at S589 destabilizes PER specifically in the nucleus. As a side note, S589D is more stable than wild type PER in the cytoplasm. These results, coupled with our in vivo data, suggest that phosphorylation of S589 is required to protect PER protein from DBT-mediated degradation in the nucleus. Our data further suggest that phosphorylation of S589 is required for stabilization of PER in the presence of TIM.

## S589 substitutions differently affect circadian transcription activity

The role of S589 in protein stability is somewhat unexpected considering our genetic data showing an interaction between S589 mutants with the *dbt*[S] but not *dbt*[L] mutation (*Figure 1D*). While the *dbt*[L] mutant is defective in regulating PER stability, *dbt*[S] is not (*Kloss et al., 1998*; *Price et al., 1998*; *Syed et al., 2011*). We hypothesized that this apparent paradox could be resolved if the same residue (S589) was involved with a second, unknown function of DBT in regulating PER, in addition to its function in regulating PER stability; DBT may also affect PER inhibitory activity. To test whether DBT sites on PER affect transcriptional inhibitory activity, we used an in vivo luciferase reporter assay for PER activity (*Stanewsky et al., 1997*). Flies expressing the luciferase reporter driven by the *per* minimal promoter and the indicated *per* transgene in a *per*[0] genetic background were monitored for bioluminescence (*Figure 4A*). Both S589A and S589D mutants have shorter total

periods than control flies expressing the wild type *per* transgene. Thus, the period of bioluminescence oscillation correlates well with behavioral period (*Figures 4B* and *1C*).

To determine whether phosphomutations of the S589 residue in PER affect PER-mediated inhibition of dCLK transcription, we analyzed the two phases of bioluminescence oscillation. A single period of bioluminescence oscillation can be separated into a dCLK active phase (crest) or a dCLK inactive phase (trough). This analysis revealed that the dCLK active phase of S589A mutants is shorter than both wild type and S589D, which are equivalent (*Figure 4A and B*). On the other hand, the dCLK inactive phase of S589A and S589D mutants are equivalent to each other and shorter than

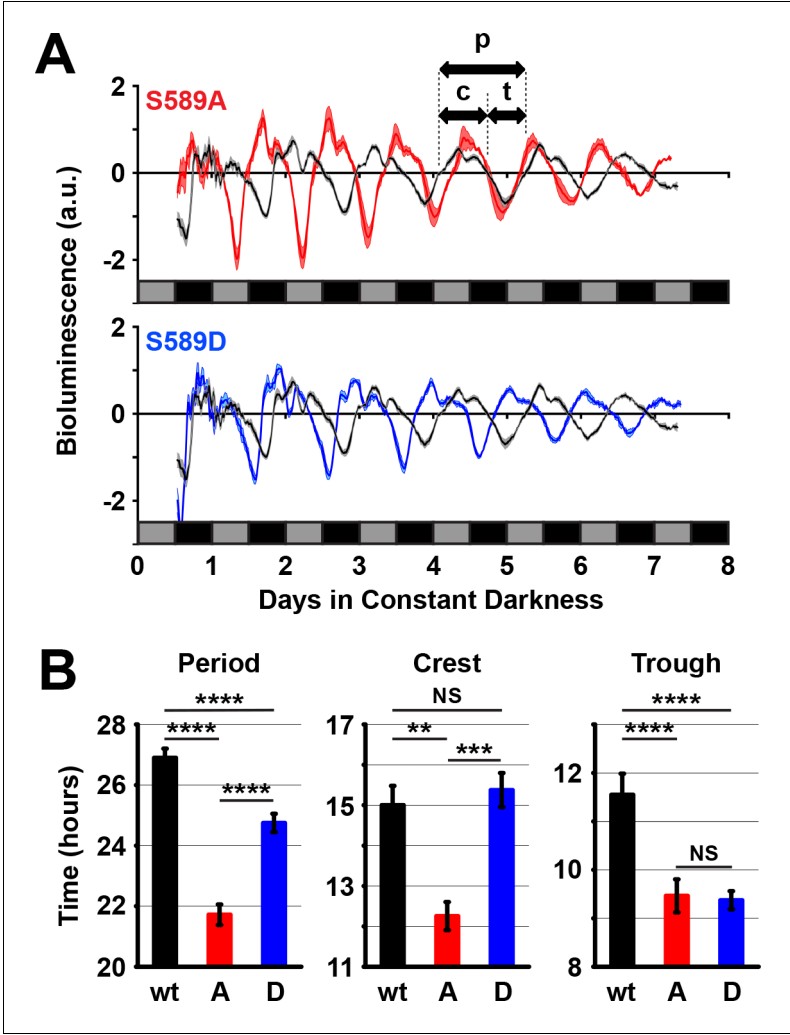

**Figure 4.** 589 substitutions differently regulate *per* promoter activity in vivo. (**A**) Flies expressing the indicated transgenic *per* variant in a *per⁰* background and the luciferase gene driven by the *per* promoter were measured for bioluminescence activity in constant darkness. Grey boxes represent subjective day, black boxes represent subjective night. S589A (red line) and S589D (blue line) were compared to wt *per* rescue (black line) flies. N = 5 groups of 80 flies. Thickness of the curve reflects SEM. Double-headed arrows use the wild type signal to illustrate a full period of bioluminescence (p), the width of the crest (c) or the width of the trough (t). (**B**) The full period of bioluminescence, and the crest and trough widths measured in panel A are plotted. w: wild type PER (black bars), A: S589A (red bars), D: S589D (blue bars). Bars represent means, ±SEM. **p<0.01. ***p<0.005. ****p<0.001. NS: not statistically significant. Significance was measured by t-test.

DOI: https://doi.org/10.7554/eLife.32679.009

The following figure supplement is available for figure 4:

**Figure supplement 1.** Realignment of bioluminescence period.
DOI: https://doi.org/10.7554/eLife.32679.010

wild type (*Figure 4A and B*). These data can also be visualized when each cycle of bioluminescence oscillation is aligned with wild type (*Figure 4—figure supplement 1*). These data suggest that the dCLK inactive phase is shortened in both S589A and S589D mutants, consistent with the role of this residue in regulating PER stability, since early PER degradation allows early dCLK activation. These data further demonstrate that the dCLK active phase is shortened only in the S589A mutant, suggesting that inhibition of phosphorylation at S589 specifically accelerates PER activity (dCLK inhibition). In other words, these data suggest that while both S589A and S589D mutants exhibit shorter circadian periods because of their effects on PER stability, S589A mutants exhibit a much shorter circadian period than S589D partly because inhibition of phosphorylation at this residue increases or accelerates PER-mediated inhibition of dCLK transcription.

## Distinguishing PER inhibitory activity from PER stability

The above data suggest that inhibitory activity of PER is increased in S589A mutants while stability is decreased. On the other hand, phosphorylation of PER is correlated with increased PER-mediated inhibition and decreased PER stability, but the two activities are difficult to assess independently (*Edery et al., 1994*; *Kivimäe et al., 2008*). To distinguish between PER activity and stability as a function of phosphorylation state, we developed a novel flow cytometry based transcription assay in S2 cells to monitor both functions independently using *per* mutants that represent transitional isoforms informed by the phosphorylation cascade model (*Figure 5*). To monitor PER inhibition of dCLK activity, we measured expression of *yfp* fused to a PEST degradation sequence driven by a *per* promoter, which is activated by dCLK; *clk* itself was CFP tagged and built into the same vector so we could monitor dCLK expression. To monitor PER expression, we co-transfected this dual-expression vector with a vector that constitutively expresses *per* fused to mCherry. Using this system, we can measure PER stability (mCherry fluorescence) simultaneously with PER inhibitory activity (YFP fluorescence) (*Figure 5A*). Co-transfection of the two vectors allowed for variability between dCLK and PER expression, but not dCLK and YFP expression. Thus, a low YFP signal can be definitively attributed to PER-mediated inhibition of transcription. Cells were imaged to verify expression and transcriptional inhibition of dCLK in PER-expressing cells (*Figure 5B*). dCLK expressing cells were identified and quantified by flow cytometry for CFP, YFP and mCherry expression (*Figure 5—figure supplement 1A*). Cells were then binned across the axis representing the mCherry fluorescence signal. Within each bin, YFP signal was normalized to dCLK (YFP/CFP) and plotted as a function of PER signal normalized to dCLK (mCherry/CFP) to measure PER activity and PER expression. The average fluorescence signal across all cells from the flow cytometry data correlates well with PER levels measured by immunoblotting (*Figure 5—figure supplement 1B–C*). Using this assay, we confirmed the expected inverse correlation between YFP expression and wild type PER signals (*Figure 5C*, open black circles).

To assay the effects of S589 mutation on PER activity and stability, we compared PER variants containing S589A or S589D with wild-type PER. We found that PER protein containing the S589A phosphonull mutation exhibits a steeper negative slope than wt PER, suggesting that S589A mutant PER is a more effective inhibitor (*Figure 5C*, red circles). Consistent with our in vivo data, S589D PER exhibits a slope statistically indistinguishable from wild type (*Figure 5C*, blue circles). The extension of the curve across the x-axis is a measure of PER stability, demonstrating higher concentration of PER in individual cells, which can be quantified by plotting the percentage of cells as a function of PER (mCherry) signal (*Figure 5—figure supplement 1B*). The destabilizing effect of mutating S589, and the relative stability of S589D over S589A is in agreement with the stability exhibited by these mutants in vivo (*Figure 2C and D*, *Figure 5—figure supplement 1B–1C*). Our data also demonstrate that S589A and S589D mutants inhibit dCLK-mediated transcription, with stronger transcriptional inhibition demonstrated by S589A. The relative instability of S589D compared to wild type PER seen here but not in vivo may be due to the inability of dCLK-mediated endogenous *tim* expression to stabilize high levels of actin-driven exogenous *per* expression. Differences in inhibitory activity or stability are not due to poor nuclear accumulation (*Figure 5—figure supplement 1D*), which is facilitated by sufficient TIM expression mediated by dCLK activity in S2 cells (*Saez et al., 2007*). This transcriptional inhibition assay therefore allows us to distinguish between mutations that affect transcription, protein stability or both. Our data demonstrate that both S589A and S589D mutants exhibit decreased stability relative to wt PER, while the S589A mutant exhibits increased PER activity relative to both wt PER and S589D. Thus, the phosphorylation state of S589 delays the circadian

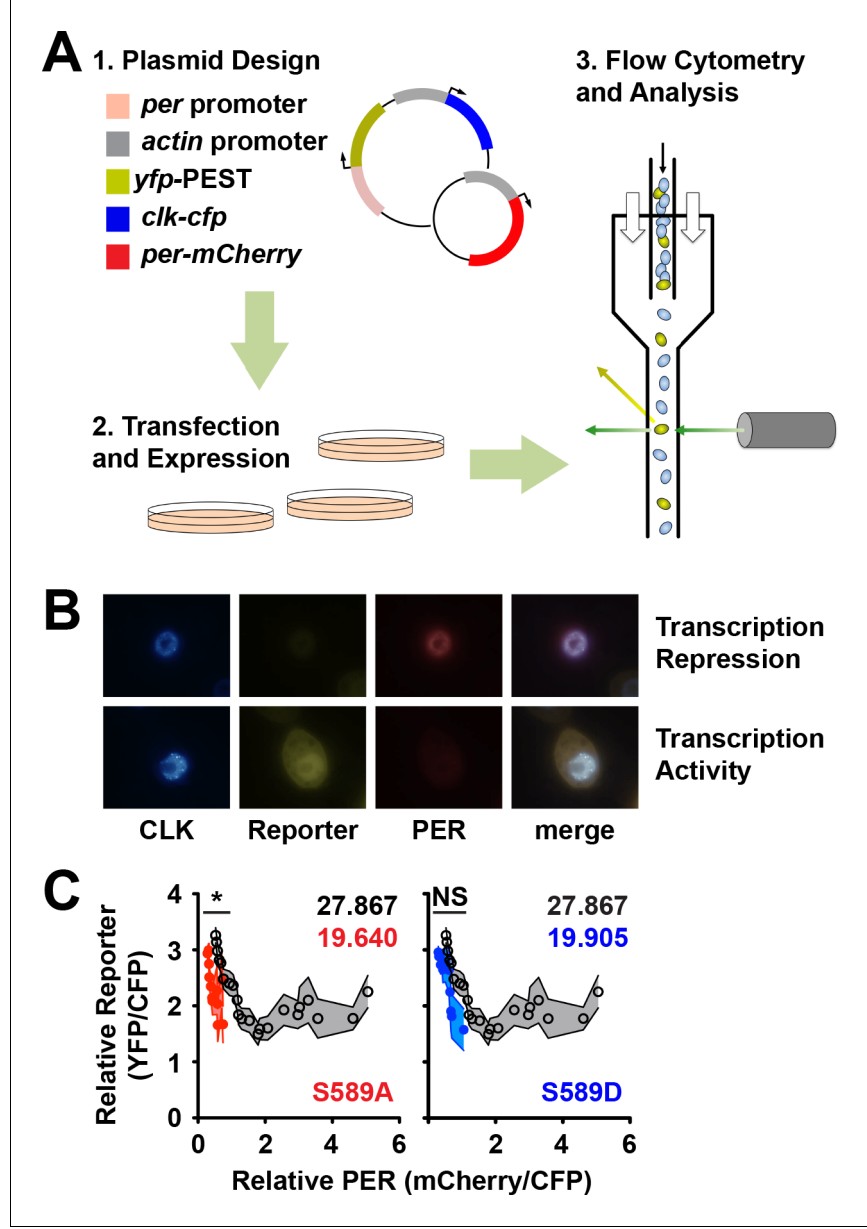

**Figure 5.** Fluorescence based transcription assay allows simultaneous measurement of transcription inhibition and protein stability. (**A**) Flow chart of the principles behind the fluorescence-based transcription assay. (1) Actin promoter (grey) driven *clk-cfp* (blue) and *per* promoter (light red) driven *yfp* (yellow) is cloned into the same expression plasmid and co-expressed in S2 cells with plasmid carrying actin promoter (grey) driven *per-mCherry* (red). (2) Cells are transfected and then (3) analyzed for fluorescence signal by flow cytometry. (**B**) Representative images of S2 cells expressing plasmids described in panel A that express high PER and low reporter (top row) and low PER and high reporter (bottom row). Expression of dCLK (blue), reporter (yellow), PER (red) is shown, with a merged image of all three signals. (**C**) CFP (dCLK)-positive cells were quantified for YFP (reporter) and mCherry (PER) expression using flow cytometry. mCherry signal was binned, and both mCherry and YFP signals were normalized to CFP signal and plotted as shown. S589A (red dots) and S589D (blue dots) were compared to wild-type PER (open black circles). Each data point represents the mean of the binned cells, ±SD. Thickness of the curve reflects SD. *p<0.05. NS, not statistically significant. Significance was measured by Mann-Whitney test. Colored inset numbers represent corresponding PER fluorescence signal per cell in arbitrary units as a measure of PER stability (Black numbers represent wild type PER stability).

DOI: https://doi.org/10.7554/eLife.32679.011

The following figure supplement is available for figure 5:

**Figure supplement 1.** Protein stability and nuclear accumulation in the transcription inhibition assay.

*Figure 5 continued*

DOI: https://doi.org/10.7554/eLife.32679.012

transcriptional feedback loop in two ways: by stabilizing PER and delaying its degradation and by regulating PER-mediated inhibition of dCLK transcriptional activity.

## Distinct phospho-programs govern PER degradation and inhibitory activity

PER stability is proposed to be regulated by the DBT phosphorylation of PER in a cascade that culminates in PER S47 phosphorylation, ubiquitination and degradation, which is antagonized by S589 phosphorylation (*Figure 6A*) (*Chiu et al., 2011*; *Garbe et al., 2013*; *Kivimäe et al., 2008*). Blocking S589 phosphorylation promotes PER degradation (*Figure 3*), suggesting that a second phosphorylation route that bypasses S589 triggers PER degradation (Routes I and II). PER-SD phosphorylation that promotes S589 phosphorylation can independently promote S47 phosphorylation and PER degradation (*Chiu et al., 2011*; *Garbe et al., 2013*). Thus, PER degradation through S47 can be regulated through two routes of phosphorylation, diverging with S589 phosphorylation. Why two routes exist to promote PER degradation through the same mechanism (S47 phosphorylation) has not been addressed. It is possible that selection of either of the two routes of PER depends on the timed regulation of PER inhibitory activity or PER degradation. We therefore set out to determine the effect of promoting the activation of each phosphorylation route and determining its effect on PER stability and activity.

To test the functional relevance of the predicted phospho-states of PER regulated by DBT, we first tested the function of the PER-SD sites (S604, S607, T610, S613) that precede S589 and S47 phosphorylation (*Garbe et al., 2013*; *Kivimäe et al., 2008*). When these sites are substituted with alanines to block phosphorylation (PER 4A), transcription is weakly inhibited (*Figure 6B*), consistent with the long-period behavioral phenotype seen in similarly mutated flies (*Garbe et al., 2013*). This protein also exhibits increased stability in comparison to wild type, producing relatively large amounts of PER (*Figure 6B*, *Figure 5—figure supplement 1B*), again consistent with delayed PER degradation in the s-LNvs (master pacemaker neurons) of similarly mutated flies (*Garbe et al., 2013*). Phosphomimetic substitutions at the same sites (PER 4D) exhibit wild type-like inhibitory activity, consistent with wild type behavioral rhythmicity seen in similarly mutated flies (D. Garbe, personal communication, December 2013), but also exhibit decreased stability in our transcription assay (*Figure 6B*, *Figure 5—figure supplement 1B*). Thus in our assay, blocking phosphorylation of the PER-SD delays PER degradation, while mimicking phosphorylation in this region reduces PER stability. Functionally, mutations that block phosphorylation of the PER-SD exhibit lower transcriptional inhibitory activity, while phosphomimetic mutations are indistinguishable from wild type.

The PER-SD region can also regulate PER stability and activity by blocking S589 phosphorylation (*Figure 6A*). To determine how the PER-SD and S589 cooperate to regulate PER stability, we generated mutants that stabilize predicted transitional phospho-states of PER and that avoid endogenous phosphorylation events, and monitored their activity and stability (*Figure 6C*). The phosphorylation-blocked mutant form (S589A/4A) results in a stable protein that is a poor inhibitor of dCLK-mediated transcription, similar to what is observed when a small DBT binding site on PER is deleted (*Kim et al., 2007*). Thus the increased inhibitory activity of S589A is muted when combined with PER 4A. In S2 cells, the PER-SD region is phosphorylated without exogenous DBT or TIM expression (*Chiu et al., 2008*), suggesting that PER-SD occurs before S589 phosphorylation, and is independent of TIM binding. The first stage of phosphorylation represented by S589A/4D reveals a strong, unstable inhibitor that resembles S589A. Thus, PER phosphorylation blocked at S589 is likely re-directed through Route I for degradation (*Figure 6A*), while still maintaining its inhibitory activity established by a phosphorylated PER-SD. To determine the consequence of permitting S589 phosphorylation, we substituted both S589 and the PER-SD serines with aspartates (S589D/4D) to reveal a PER variant that is a strong and stable inhibitor. Analysis of purified PER fragments by size exclusion chromatography multi-angle light scattering (SEC-MALS) and small-angle X-ray scattering (SAXS) revealed no change in the global structure of either protein, with overall shapes of the proteins approximately the same, and with little difference in their flexibility as determined by trypsin protection assay.

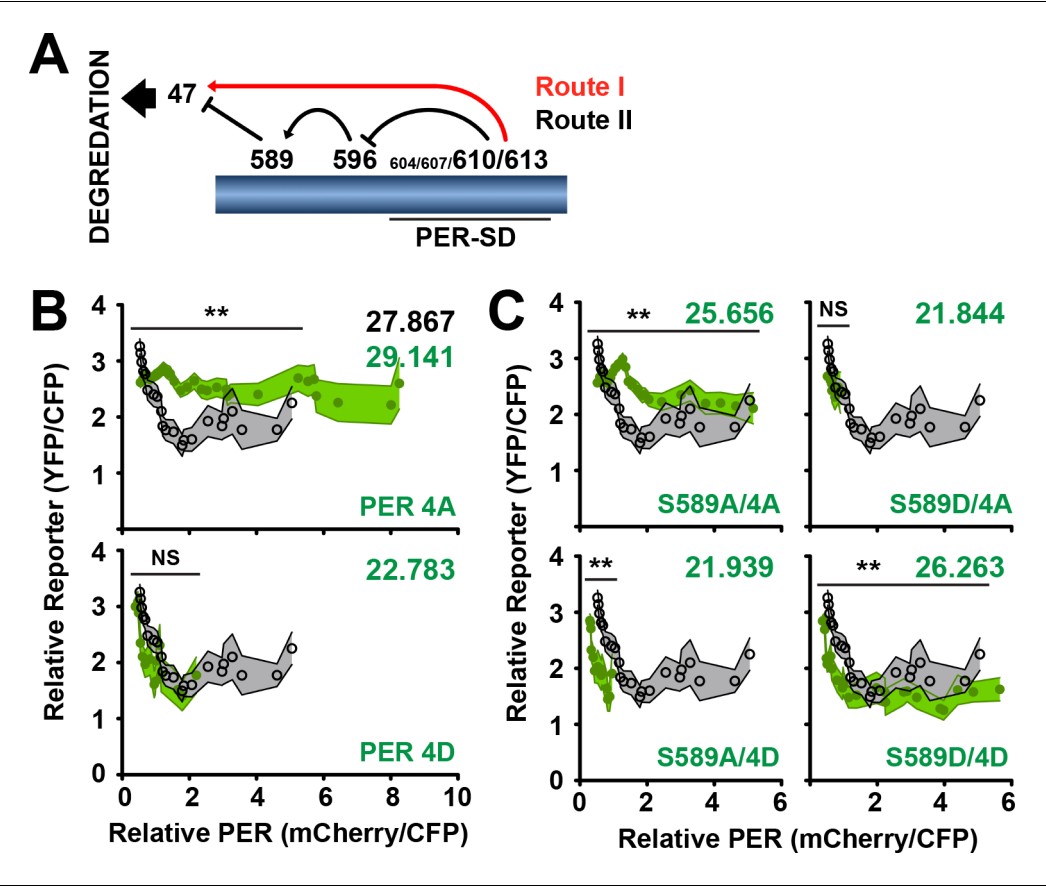

**Figure 6.** S589 and the PER Short Downstream Domain (PER-SD) cooperate to regulate transcriptional inhibition and PER protein stability. (A) The phosphorylation cascade model of PER. Phosphorylation of the PER-SD region blocks S589 phosphorylation through an intermediary site (S596) while promoting S47 phosphorylation. S589 phosphorylation blocks S47 phosphorylation. S47 phosphorylation promotes PER degradation. Routes I and II are labeled as they are referred to in the text. (B) The transcription analysis described in *Figure 5* is applied to *per* mutants PER 4A (S604A/S607A/T610A/S613A) and PER 4D (S604D/S607D/T610D/S613D) (green circles) and compared to wild type (open black circles). Each data point represents the mean of the binned cells, ±SD. Thickness of the curve reflects SD. **p<0.01. NS, not statistically significant. Significance was measured by Mann-Whitney test. Colored inset numbers represent corresponding PER fluorescence signal per cell in arbitrary units as a measure of PER stability (Black numbers represent wild type PER stability). (C) The transcription analysis is applied to *per* mutants S589A/4A, S589A/4D, S589D/4A, S589D/4D (green circles) and compared to wild type (open black circles). Each data point represents the mean of the binned cells, ±SD. Thickness of the curve reflects SD. **p<0.01. NS, not statistically significant. Significance was measured by Mann-Whitney test.

DOI: https://doi.org/10.7554/eLife.32679.013

The following figure supplement is available for figure 6:

**Figure supplement 1.** Residue substitutions of DBT target sites do not affect global conformational differences in purified PER fragment, but affect complex assembly.

DOI: https://doi.org/10.7554/eLife.32679.014

These data suggest that phosphorylation does not affect the global conformation of PER but may alter local structure in a manner that influences its function (*Figure 6—figure supplement 1*). Thus, phosphorylation of S589 commits PER to Route II, leading to a PER isoform that is stable and active through subtle changes to protein conformation. S589 phosphorylation is likely not blocked by PER-SD phosphorylation, but functions as a switch in delayed transcription in the circadian clock, committing PER to degradation or stable inhibition.

A PER variant not predicted by the phosphorylation model, S589D/4A, is an unstable protein with indeterminate inhibitory activity that resembles S589D (*Figures 5C* and *6C*). This transition state can occur if S589 phosphorylation blocks PER-SD phosphorylation (*Fu, 2008*; *Kivimäe et al., 2008*) and may represent (i) a later stage isoform of PER at the end of transcriptional inhibition, (ii) an isoform that commits PER to degradation prematurely, or (iii) may not be an isoform relevant to the PER degradation pathway.

## Discussion

Delays in the transcriptional negative feedback loop of the circadian clock ensure that it oscillates with ~24 hr periodicity. Delayed transcription (i.e. delayed PER degradation and transcription re-activation) is one such critical delay. In this study, we present a model in which DBT stabilizes and activates PER inhibition to delay transcription in the circadian clock (*Figure 7*). DBT begins phosphorylation of PER in the PER-SD (*Garbe et al., 2013*; *Kivimäe et al., 2008*) (Step 1), initiating inhibition of dCLK-mediated transcription by PER. S589 is subsequently phosphorylated to stabilize the PER/TIM inhibitory complex in its inhibition-active form (Step 2). Although protein extracts from whole heads suggest that S589 is phosphorylated at ZT20, after nuclear entry (*Chiu et al., 2011*), the stabilization effect S589 phosphorylation has on PER is apparent in the s-LNvs after CT02 (*Figure 2*), and regulates the temporal delay of transcription reactivation. Upon TIM degradation, PER is released from TIM protection and is degraded after CT06 (Step 3). Thus, the steps that delay dCLK-mediated transcription activity rely on the activation and stabilization of PER by DBT phosphorylation, before TIM degradation ensues.

DBT is likely to regulate the stability and inhibitory activity of PER through separate mechanisms. Such functional uncoupling has also been suggested for the Neurospora clock (*Larrondo et al., 2015*). Regions near the N- and C-termini of PER regulate PER stability and inhibitory activity, respectively. Phosphorylation of S47 at the N-terminus of PER regulates PER-SLIMB binding that leads to the ubiquitination and degradation of PER (*Chiu et al., 2008*). The dCLK/CYC Inhibitory Domain (CCID) at the C-terminus of PER physically interacts with dCLK to block transcriptional activity (*Chang and Reppert, 2003*). These two functional regions of PER correlate with the two predicted DBT binding sites that reside at the PER N-terminus (residues 1–365) and a third of the way from the C-terminus (residues 755–809) (*Kim et al., 2007*; *Kloss et al., 1998*). The two target regions of DBT also provide an explanation for the function of the long- and short-period mutant forms of DBT. Both long-period (*dbt^L*) and short-period (*dbt^S*) behavioral phenotypes carry mutations that reduce the kinase activity of DBT (*Kivimäe et al., 2008*; *Preuss et al., 2004*; *Price et al., 1998*). Lowered kinase activity that produces divergent behavioral phenotypes are explained by

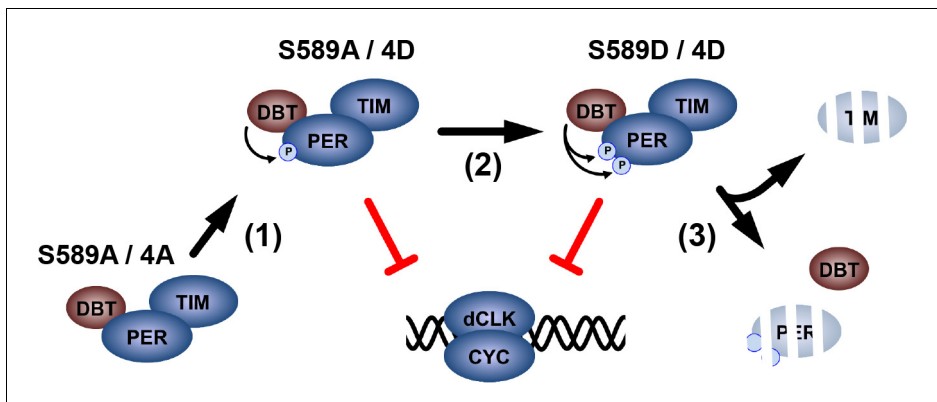

**Figure 7.** A proposed model for DBT-mediated PER regulation. Non-phosphorylated PER is stable, but unable to inhibit dCLK-mediated transcription. DBT binds PER and phosphorylates the PER-SD to activate PER inhibition of dCLK (Step 1). DBT phosphorylates S589 stabilizing the PER/TIM complex, while PER continues to inhibit dCLK (Step 2). Subsequently, TIM is degraded, releasing PER for degradation, ending the PER life cycle.
DOI: https://doi.org/10.7554/eLife.32679.015

likely changes in DBT activity on specific PER target sequences. Indeed, we show that the S589 mutant flies resistant to the effects of a $dbt^S$ mutant background have periods that are lengthened in a $dbt^L$ mutant background (*Figure 1D*). These data suggest that $dbt^S$ genetically interacts with S589, as inferred by others (*Rothenfluh et al., 2000*), while $dbt^L$ does not. Since $dbt^L$ and not $dbt^S$ delays PER degradation in cultured S2 cells (*Syed et al., 2011*), $dbt^S$ may affect PER inhibition function through its (in)activity in the S589/PER-SD region.

Distinct phospho-programs also regulate human PER2 stability and inhibitor activity (*Xu et al., 2007*). The human study focuses on the hPER2 variants S662G and S662D, analogous to the PER S589A and S589D variants in flies, targeted by the mammalian orthologue of DBT, CKIδ. Unlike S589D, S662D exhibits a long behavioral period. This difference may be due to the differences in how these sites regulate PER. Although both S589A and S662G lead to a reduction of PER or hPER2 respectively, the decreased levels of the hPER2 S662G mutant is primarily due to increased hPER2 inhibitory activity at the *hper2* locus (decrease in transcription), rather than a change in protein stability. On the other hand, S589 phosphorylation (S589D) stabilizes PER in complex with TIM and helps activate PER inhibitory activity. Despite these differences, both S589A and S662G advance evening anticipation behavior, suggesting that phospho-regulation of these sites underlie this aspect of rhythmic behavior. Also, a reduction of DBT activity in mutant flies ($dbt^L$) or reduced CKIδ expression in mice (heterozygous expression) both lead to longer period behavioral rhythms that are independent of residue substitution in positions S589/S662. Thus, despite differences in the PER function that sites S662 and S589 regulate, both mammals and flies utilize distinct mechanisms that employ DBT/CKIδ to separately regulate the activity and stability of Period protein, delaying transcription reactivation in the circadian clock.

The relationship between the DBT phosphorylation cascade, PER inhibitory activity and PER stability is not well understood. *In vivo*, it is difficult to determine if a mutation in *per* results in lower PER protein levels caused by destabilized protein or overactive transcriptional inhibition. To distinguish between the two possibilities, we must simultaneously measure transcription factor expression and activity. We therefore developed a fluorescence-based transcription assay to probe the relationship between S589 and the PER-SD in regulating stability and inhibitor activity. The previous PER phosphorylation model begins with phosphorylation of PER-SD that blocks PER S589 phosphorylation, and promotes PER S47 phosphorylation (*Chiu et al., 2011*; *Garbe et al., 2013*) (*Figure 6A*). Residue substitutions in PER that block and mimic phosphorylation reveal a step-wise activation and stabilization of PER in our transcription assay (*Figure 6*). S589A/4D represents the initiation of the phosphorylation cascade that activates PER transcriptional inhibitory activity. This isoform is unstable however, stabilized in inhibitor-active form with the phosphorylation of S589, in the form of S589D/4D. This suggests no antagonistic relationship between the two regions. The previous model also predicts that S589 cannot be phosphorylated when the PER-SD region is (*Chiu et al., 2011*; *Garbe et al., 2013*). If this were the case, S589A/4D and PER 4D would behave similarly. Instead, S589A/4D appears more like S589A and PER4D more like S589D/4D in our assay, again suggesting no antagonistic relationship between the two regions. In vivo, S589 is phosphorylated at ~ZT20, and appears to remain phosphorylated as PER is degraded (*Chiu et al., 2011*). Other data suggest that the PER-SD region is phosphorylated before S589, based on mass spectrometry of PER isolated from cultured cells (*Chiu et al., 2008*), or epigenetic data (*Garbe et al., 2013*). Thus, it is not clear that the isoform represented by S589A/4D is a long-lived isoform of PER, assuming that the PER-SD region is indeed phosphorylated prior to S589. If this were true, it would suggest that this unstable early isoform of PER (mimicked by S589A/4D) requires rapid stabilization. There is no apparent need for an unstable PER protein early in transcriptional inhibition and it is more likely that both the S589 and PER-SD sites cooperate to regulate PER stability and function. Therefore, we suggest that PER-SD promotes S589 phosphorylation to stabilize and activate PER inhibition, and delay transcription by inhibiting dCLK, which peaks around CT02. Indeed, S589 phosphorylation is relevant after CT02 to stabilize the PER/TIM complex (*Figure 2D and E*).

Phosphatases play a role in maintaining 24 hr behavioral rhythms (*Chen et al., 2007*; *Fang et al., 2007*). If phosphatases act on the stable inhibitor represented by S589D/4D, they can convert this PER variant to either an isoform represented by S589A/4D, or through multiple actions, to the S589D/4A isoform. In such a scenario, the re-formed S589A/4D representative isoform can be rephosphorylated, pushing the clock forward. On the other hand, the S589D/4A representative isoform would be rapidly degraded, disallowing the clock from shifting backward. The small number of

S589D/4A variants that escape degradation and become dephosphorylated to form S589A/4A can be re-phosphorylated through the same phosphorylation cascade. Thus the ordered phosphorylation of PER-SD and then S589 ensures that the circadian clock ticks forward.

Mutations that block regulatory mechanisms of the circadian clock can cause significant health problems. We have shown that DBT-mediated activation, stabilization and degradation of PER regulates delayed transcription during the day. Mutations that interfere with this delay mechanism shorten perceived day and hasten evening anticipation (*Figure 1A*). Thus, we demonstrate a biochemical mechanism that influences one aspect of circadian behavior (evening anticipation). Similarly, a single mutation that alters splicing of *cryptochrome* (transcriptional inhibitor in humans) alters its binding affinity to Clock, also changing the delay of transcription inhibition in humans to cause a familial form of delayed sleep phase disorder (*Patke et al., 2017*).

## Materials and methods

### Plasmids, tissue culture, and transfection

DNA encoding *tim*, *per*, *dbt*, *clk* (and the indicated corresponding mutant) was subcloned into the HS-Casper (*Saez and Young, 1996*) or pAc5.1/V5-HisA plasmids (Invitrogen, Norwalk, CT). Mutant *per* genes were subcloned into each vector using standard cloning techniques. CFP, YFP, mCherry, 3xHA, 3xmyc, 3xFLAG, 2xV5 were fused to the C-termini of the indicated genes, separated by a triple-glycine linker. Plasmid expressing *yfp* fused to a PEST sequence by *per* promoter (per > yfp-PEST) was a gift from Sheyum Syed. A HindIII site 3' to the YFP-PEST cassette was used to insert a pAc > *clk*-cfp cassette to generate the reporter plasmid. S2 cells were obtained from ATCC (Manassas, VA) maintained in Schneider's medium (Invitrogen) supplemented with 10% FBS (Sigma-Aldrich, St. Louis, MO) and tested negative for mycoplasma contamination. S2 cells were transfected using Effectene (Qiagen, Hilden, Germany) as per manufacturer's protocols. Cells were resuspended in fresh medium one day after transfection and seeded onto Lab-Tek II chamber slides (Nunc, Rochester, NY) for imaging.

### Transgenic flies and behavioural analysis

The *per* mini-gene (*Bargiello et al., 1984*) was mutated using standard cloning techniques, and subcloned into an attB plasmid (*Bischof et al., 2007*) with a C-terminal 3xmyc tag. Plasmids were injected into attp40 *Drosophila* lines to generate transgenic lines (BestGene, Chino Hills, CA). Individual flies were analyzed for locomotor activity in 12 hr: 12 hr light/dark light regiments (LD) or in constant darkness (DD) after at least three days entrainment, using the *Drosophila* Activity Monitor System IV (Trikinetics, Waltham, MA). Period was determined using ClockLab Software (Actimetrics, Wilmette, IL). UAS-DBT transgenic flies were a gift from Jeffrey Price (University of Missouri, Kansas City). UAS-OGT transgenic flies were a gift from Louis Ptáček (University of California, San Francisco). Flies overexpressing OGT were generated by recombining the second chromosome carrying the tim-UAS-Gal4 and UAS-OGT transgenes.

### Luciferase assay

Flies expressing the luciferase gene by *per* promoter (*plo*) were crossed to flies expressing the indicated *per* transgenes in a *per*⁰ background to generate a y,w,*per*⁰;plo/[*per*];+genotype for analysis. Two sets of 80 males were monitored in 3 cm plates with standard fly food supplemented with luciferin (Cayman Chemicals, Ann Arbor, MI), as previously described (*Stanewsky et al., 1997*) in a scintillation counter (Hamamatsu Photonics, Model LM2400, Hamamatsu, Japan). Bioluminescence was measured in photons/10 min.

### Immunofluorescence microscopy

Fly brains were collected, fixed, mounted, and imaged using Leica confocal microscopy as previously described (*Saez et al., 2011*; *Top et al., 2016*). Briefly, fly heads were fixed in PBS with 4% paraformaldehyde and 0.5% Triton X-100. Brains were dissected and washed in PBS with 0.5% Triton X-100 and blocked in the same solution supplemented with 5% donkey serum. Brains were then probed with 1:200 dilution of PER antibody, or 1:1000 dilution of PDF antibody (DSHB, Iowa City, IA). Washed brains were re-probed using 1:200 diluted secondary antibodies conjugated to Alexa-488

(PER) or Alexa-647 (PDF). Brains were mounted using Fluoromount (Beckman Coulter, Brea, CA) and imaged using Leica confocal microscopy at 40X magnification. Fluorescence intensity was quantified using ImageJ, with bright and dark reference points to maintain consistent relative quantification across all images. In quantifying protein expression, entire cells were analyzed with no distinction between nuclear and cytoplasmic signal.

## Fluorescence microscopy of S2 cells

To measure nuclear accumulation, S2 cells in chamber slides (see above) were heat shock induced to exogenously express TIM (YFP) or PER (mCherry) alongside actin-driven PER (mCherry) or TIM (YFP), respectively, and actin-driven CFP in an air incubator at 37°C for 30 min. Cells were imaged using a DeltaVision system (Applied Precision, Issaquah, WA) equipped with an inverted Olympus IX70 microscope (60X oil objective, 1.42 N.A.), a CFP/YFP/mCherry filter set and dichroic mirror (Chroma, Foothill Ranch, CA), a CCD camera (Photometrics, Tucson, AZ), and an XYZ piezoelectric stage for locating and revisiting multiple cells. 50–70 cells were selected based on their CFP expression and CFP, YFP, mCherry channels recorded. Imaged cells were analyzed using a locally written algorithm in Matlab (Mathworks Inc, Natick, MA), as previously described (*Top et al., 2016*). To measure PER inhibition of YFP reporter, S2 cells expressing the dCLK/YFP reporter plasmid and PER were seeded on chamber slides and imaged similarly, with no heat shock treatment. 10–20 cells were selected based on their CFP (dCLK) expression and CFP, YFP, mCherry channels recorded and typical results shown.

## Pulse-Chase assay and immunoblotting

S2 cells were transfected in 10 cm dishes with plasmids expressing the indicated genes as described above. At the end of two days, cells were resuspended in 11 ml of medium and placed in 15 ml conical tubes. The tubes were heat shocked in a 37°C on a nutator for 30 min, and recovered for another 30 min at room temperature. Samples (1 ml) were taken at the indicated time points (once an hour), with cycloheximide (20 µg/ml) added at hour 1 (1 hour post heat shock recovery). Cells were washed once with PBS and lysed using 200 µl modified RIPA (50 mM Tris pH 8.0, 150 mM NaCl, 1 mM EDTA, 1% Triton X-100, 0.5% Na-Deoxycholate, 25 mM NaF, 1 mM DTT, 20% Glycerol, 0.01% $NaN_3$). Equal volumes (15 µl,~5–10 ug total protein) of lysate were used in analysis by immunoblot using 6% or 10% SDS-PAGE gels. Blots were probed with 1:1000 anti-TIM antibody (*Myers et al., 1996*), 1:1000 anti-PER antibody (*Top et al., 2016*), 1:10,000 anti-V5 antibody (Sigma-Aldrich), 1:3000 anti-HA antibody (Roche, Basil, Switzerland), 1:10,000 anti-myc antibody (Sigma-Aldrich), 1:10,000 anti-tubulin antibody (Sigma-Aldrich). Protein bands were quantified using imageJ freeware.

## Flow cytometry

S2 cells expressing the reporter/*clk-cfp* plasmid described above were analyzed on a BD LSR-II cell sorter (BD Biosciences, Franklin Lakes, NJ) using FACSDiva v.6.1.1 software. Fluorescence signal was filtered to monitor CFP (405 nm), YFP (488 nm) and mCherry (561 nm). CFP positive cells (N =~10,000) were binned into groups of increasing mCherry signal and plotted as a function of YFP signal.

## Protein purification and SEC-MALS analysis

Variants of *per* (amino acids 1–700) were sub-cloned into pGEX-6–1 p plasmid using standard cloning techniques. Bacteria were induced to express GST-fused PER fragment at 17°C at $OD_{600}$ 0.6 and grown overnight. Bacteria were lysed chemically in lysis buffer (500 mM NaCl, 50 mM Tris pH 7.5, 5 mM DTT, Bug Buster (EMD Millipore, Billerica, MA), RNase, and benzonase nuclease) supplemented with protease inhibitors. PER fragments were purified using glutathione beads affinity chromatography, the GST tag was cleaved by prescission protease (GE Lifesciences, Pittsburgh, PA), followed by size exclusion chromatography purification, and confirmed by mass spectrometry. SEC-MALS was conducted at room temperature on a Wyatt–WTC050N5 SEC column with buffer containing 500 mM NaCl, 50 mM Tris pH 7.5, and 3 mM DTT.

## Small angle X-ray scattering

SAXS data was collected in a similar fashion as previously described (*Skou et al., 2014*), but with an SEC component upstream from SAXS data collection. Briefly, SEC-SAXS data was collected on a size-exclusion column pre-equilibrated with 500 mM NaCl, 50 mM Tris pH 7.5, and 3 mM DTT at the G1 line at the Cornell High Energy Synchrotron Source (CHESS) on a dual 100K PILATUS detector system. Approximately 10 mg/ml PER variants were injected onto the SEC column at a continuous flow rate of 0.2 ml/min. Data was collected with exposures of 2 s per frame ($5 \times 10^{11}$ photons/sec). Blank buffer with no protein was exposed and collected as background for subtraction. Lysozyme and glucose isomerase were used as standards for Rg and molecular weight calculations. RAW (*Nielsen et al., 2009*) and Primus (*Konarev et al., 2003*) were used to generate plots in *Figure 6—figure supplement 1*. 'q' is calculated as $q = 4\pi\sin(\theta)/\lambda$, where $\theta$ is half of the angle between the incident X-ray beam and the scattered beam, and $\lambda$ is the wavelength of X-rays.

## Trypsin protection assay

PER fragments (2 µl; 80 µM) were incubated with trypsin (80 µM) for the indicated times, and compared to undigested controls. Digestion reactions were stopped using 2 µl of 160 µM trypsin inhibitor and 10 µl SDS gel loading buffer. Digested products were analyzed by 4–12% NuPAGE BisTris gel, Coomassie stain, and quantified using ImageJ. Protein fragments were confirmed as PER fragments by mass spectrometry.

## Acknowledgements

We would like to thank Wanhe Li for critical review of the manuscript. This work was supported by grants from the National Institutes of Health (GM054339) to MWY, and National Institutes of Health (GM079679 and GM122535) to BRC.

## Additional information

### Funding

| Funder | Grant reference number | Author |
| --- | --- | --- |
| National Institutes of Health | GM079679 | Brian R Crane |
| National Institutes of Health | GM122535 | Brian R Crane |
| National Institutes of Health | GM054339 | Michael W Young |

The funders had no role in study design, data collection and interpretation, or the decision to submit the work for publication.

### Author contributions

Deniz Top, Conceptualization, Data curation, Formal analysis, Investigation, Methodology, Writing—original draft, Project administration, Writing—review and editing; Jenna L O'Neil, Data curation, Formal analysis, Validation, Visualization, Methodology; Gregory E Merz, Kritika Dusad, Data curation, Formal analysis, Methodology, Writing—original draft; Brian R Crane, Conceptualization, Resources, Supervision, Funding acquisition, Validation, Writing—original draft; Michael W Young, Supervision, Funding acquisition, Writing—original draft, Writing—review and editing

### Author ORCIDs

Deniz Top http://orcid.org/0000-0002-1042-8460
Brian R Crane http://orcid.org/0000-0001-8234-9991

### Decision letter and Author response

Decision letter https://doi.org/10.7554/eLife.32679.020
Author response https://doi.org/10.7554/eLife.32679.021

## Additional files

### Supplementary files

• Supplementary file 1. Fly behavioral data. Behavioral periods of flies charted in *Figure 1B, C and D* are detailed. Tau: behavioral period, SD: standard deviation, No. Arr.: Number of arrhythmic flies, Total Number: Total number of flies tested. fs: figure supplement.

DOI: https://doi.org/10.7554/eLife.32679.016

• Transparent reporting form

DOI: https://doi.org/10.7554/eLife.32679.017

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
