## [Decision Letter]

Thank you for submitting your article "CK1/Doubletime Activity Delays Transcription Activation in the Circadian Clock" for consideration by *eLife*. Your article has been favorably evaluated by a Senior Editor and three reviewers, one of whom is a member of our Board of Reviewing Editors. The following individual involved in review of your submission has agreed to reveal their identity: Louis J Ptáček (Reviewer #1).

The reviewers have discussed the reviews with one another and the Reviewing Editor has drafted this decision to help you prepare a revised submission.

Summary:

In animals, PER is the main repressor of CLOCK-BMAL1/CYCLE dependent circadian transcription. PER undergoes a complex hierarchical phosphorylation program that regulates its stability in a daily manner. In *Drosophila*, numerous studies have led to key insights into how phosphorylation regulates PER stability. Key insights have been gained from the classic *per-short* mutation at Ser589 that abrogates phosphorylation at this site. The *per-s* mutation accelerates PER degradation, and TIMELESS (TIM) stabilizes PER against DBT-mediated degradation. In this paper, Top et al. show that phosphorylation at Ser589 by DBT has a dual purpose in the transcriptional feedback loop by delaying PER degradation and making it a better repressor. The transition from repression to degradation is temporally linked to TIM. The data is largely based on analyzing an ala mutant of Ser589 (non phospho form) and a novel Asp mutant (phosphomimetic?). To better measure PER stability (levels) and transcriptional repression, they devised a tissue culture based flow cytometry assay that can score both simultaneously. By incorporating prior findings, they suggest a model whereby upstream phosphorylation of PER (SD region) precedes phosphorylation at Ser589 by DBT, which in the presence of TIM, generates a strong PER repressor. Phosphorylation of Ser589 after CT2, stabilizes PER and allows it to function as a repressor. Later on in the cycle, when TIM is degraded, PER becomes unstable and degraded.

In an earlier paper, Young and co-workers (Kivimae et al., 2008) suggested that phosphorylation of PER at Ser589 and the *per-short* domain also affects PER repressor activity, making it less stable but a better repressor. Prior work (Nawathean and Rosbash, 2004) showed that DBT and CK2 increase PER repressor function, although the mechanism was not clear. The role of Ser589 on PER stability and the phosphorylation cascade, including how DBT and TIM function in the model presented (Figure 7), have been largely worked out before (Rothenfluh et al., 2000; Kivimae et al., 2008; Chiu et al., 2008, 2011; Garbe et al., 2013; Li and Rosbash, 2013). Novel insights were gained by analyzing a 589 Asp mutant as it also causes shorter rhythms (but not as short as the Ala mutant).

Essential revisions:

1) The key finding we all seem to agree on is that 589D/4D is a stable and strong repressor. This is a new variant of PER that has not previously been studied, as all prior studies used A mutants. This is a key finding that shows a role for 589 in both degradation and inhibition, with regards to the SD region. In addition to this finding, based on the 589D allele still having a shorter period then wildtype, they also make the claim that phosphorylation at 589 makes PER unstable once TIM is gone; i.e., the TIM switch. The fact that maintaining phosphorylation at 589 ALSO affects the period is interesting and novel. However, here the interpretation of what this means is not clear. The problem is that the species they have in their model for degradation of PER at the end of the cycle is actually S589D/4A. If that is the "wildtype" species for degradation (i.e., 589 is phosphorylated) then why does the 589D mutant have a shorter period then wildtype? Also, in their model they go from S589D/4D to S589D/4A for degradation of PER after TIM is lost. This would imply that it is not just the status of S589 that is key to degradation after TIM is lost but you also need to dephosphorylate the SD region. According to their mutant data in flies showing that 589D is shorter then wildtype, this should(?) imply that there is a species of wildtype PER at the end of the cycle whereby 589 is not phosphorylated and more stable than the 589D form that is phosphorylated. So, is there a species of PER whereby a 589D is less stable then a 589A that could account for the difference in period between S589D and wildtype? From Figure 6, S589D/4A is less stable then S589A/4A, so it might be possible that the S589D/4D becomes S589A/4A at some point, but a potential problem is that S589A/4A is too(?) stable-although it seems to be less stable then PER4A. Thus, the main problem I have is with the second part of the story; i.e., how does maintaining phosphorylation of 589 at the end of the cycle make it less stable then the wildtype, giving rise to a shorter period? In other words, they imply that de-phosphorylation of 589 late in the cycle is critical for delayed PER degradation. Why is that not in the model? And if 589 has to be dephosphorylated at some point, what is the PER species that is being degraded in the wildtype at the end of the cycle?

2) Also, the TIM switch argument is complicated. Basically, based on Figure 3, they say, "These data therefore suggest that S589 phosphorylation stabilizes PER in the presence of TIM from DBT-mediated degradation, either by blocking phosphorylation of other PER destabilizing sites, or in spite of phosphorylation of these other sites. In contrast, blocking S589 phosphorylation redirects DBT to promote the degradation of the PER/TIM complex, likely by phosphorylating other PER sites. Therefore, S589 acts as a switch, directing DBT to degrade PER or to stabilize PER within the PER/TIM complex."

- While all of this is true, calling it a TIM switch is a bit confusing. All it means is what we already know, in the presence of TIM the 589D species is more stable than the 589A species when other sites are also phosphorylated. Otherwise stated, both 589A and 589D species are more stable with TIM, it is just that 589A is less stable then 589D. In do not see how 589A degradation is "TIM-independent", when it is clearly shown in Figure 3 that TIM stabilizes 589A (this was also shown by Li and Rosbash, 2013, JBR, and should be cited).

To summarize:a) How does retaining S589 phosphorylation destabilize PER at the end of the cycle to get a shorter period compared to the wildtype situation as implied from the S589D mutant (i.e., at CT6 what is the wildtype form of PER that is more stable in comparison to flies that have 589D?). And if this implies that 589 has to be dephosphorylated at some point late in the cycle (so that it continues being stable and repressing for a few more hours), what is that species and why is it not shown in the model?

b) I do not fully understand the implication of calling the role of TIM vis. a vis. S589 a TIM switch when all phospho-isoforms of PER appear to be less stable without TIM in the presence of DBT. Indeed, from the model it appears that losing SD phosphorylation is required for degradation but not a change in the phosphorylation status of S589.

c) Chiu et al., 2011, showed that S589 is highly phosphorylated by ZT20, so it is not clear that there is an early species of S589A/4D, which is basically based on the Garbe et al., model. The Garbe model is based on epigenetics, so it is not a given that phosphorylation at SD blocks phosphorylation at S589. This potential discrepancy should be discussed. In any event, it was not discussed if there is a functional significance to the initial repressor being highly unstable, i.e., does it play a different role compared to the more stable 589D/4D?

Therefore, while the role of 589D/4D as a novel, stable and strong repressor is well documented and highly interesting, and an added role for 589 phosphorylation status in the degradation of PER after CT2 is also interesting, the latter part of this story and a specific role for TIM are less solid, making the latter part of the model unclear as to how it was derived based on the data.

When presenting the model in Figure 7, the authors need to present their arguments for the time course and nature of the phosphorylated species at each step, with alternatives presented if possible.

3) The authors largely ignore related work in mammalian systems showing that CKId has different effects on Per2 depending on the site of phosphorylation. The PER phosphorylation cascade downstream of the classic *per short* site likens to the phosphorylation cascade reported in mammalian Pers (Xu et al., 2007). This paper should be cited and discussion regarding similarities and differences of the reported findings with these published data should be addressed.

4) Another interesting observation are that the results with the S589A and S589D alleles don't all agree with what might be predicted regarding expected DBT phosphorylation effects (assuming 589D mimics a constitutively phosphorylated S589). It would be worth also looking at 2 additional papers: (Kaasik et al., 2013; Li et al., Cell Metabolism 2013). These papers show that the same Serines in the "cascade region" in mammals are also being O-GlcNAC modified. That is, CKId and OGT are competing for occupancy and modification of the same sites. An interesting question is whether O-GlcNAC modified vs. phosphorylated Serines in this region have different downstream effects. Has anyone looked to see whether the phosphorylation cascade region in dper is O-GlcNAC modified? While the fly and mammalian regions are not really conserved at a sequence level, the similarity of some of the effects in fly suggests they may be similar functionally. If so, then phosphorylated vs. GlcNAC modified vs. unmodified may underlie the unexpected results of S589 vs. D589.

5) There is no need for the SAXS data, etc. as it does not add anything to the Discussion.

---

## [Author Response]

Essential revisions:1) The key finding we all seem to agree on is that 589D/4D is a stable and strong repressor. This is a new variant of PER that has not previously been studied, as all prior studies used A mutants. This is a key finding that shows a role for 589 in both degradation and inhibition, with regards to the SD region. In addition to this finding, based on the 589D allele still having a shorter period then wildtype, they also make the claim that phosphorylation at 589 makes PER unstable once TIM is gone; i.e., the TIM switch. The fact that maintaining phosphorylation at 589 ALSO affects the period is interesting and novel. However, here the interpretation of what this means is not clear. The problem is that the species they have in their model for degradation of PER at the end of the cycle is actually S589D/4A. If that is the "wildtype" species for degradation (i.e., 589 is phosphorylated) then why does the 589D mutant have a shorter period then wildtype? Also, in their model they go from S589D/4D to S589D/4A for degradation of PER after TIM is lost. This would imply that it is not just the status of S589 that is key to degradation after TIM is lost but you also need to dephosphorylate the SD region. According to their mutant data in flies showing that 589D is shorter then wildtype, this should(?) imply that there is a species of wildtype PER at the end of the cycle whereby 589 is not phosphorylated and more stable than the 589D form that is phosphorylated. So, is there a species of PER whereby a 589D is less stable then a 589A that could account for the difference in period between S589D and wildtype?

We thank the reviewer for the constructive and helpful comments.

We are encouraged that the reviewers agree that the S589D/4D is a strong and stable repressor, and that S589 (with PER-SD) plays a dual role in helping repress CLK-mediated transcription and PER stability. (A) The key problem the reviewers have in this section is our placement of S589D/4A at the end of the PER life cycle. (B) Another problem the reviewers have is the emphasis on TIM as a regulator of PER stability, even in the context of the different mutant isoforms. (C) Finally, there is some confusion regarding the phospho-state of S589, why both S589D, and S589A can be less stable than wild type and the relationship of the changed stability to behavioral period.

The reviewers raise important points, which we clarify through changes in the text. We have also reviewed and revised our model in Figure 7.

A) After measuring the stability of the S589D/4A variant, we reasoned that it fit well nearer the end of the PER cycle. However, it is also possible that the S589D/4A isoform represents a transitional state in an alternate “short circuit” pathway to induce PER degradation, or may not be relevant for stability. Our model emphasizes that the S589D/4D isoform of PER is a strong and stable repressor and that DBT phosphorylation events mediate the stability of PER, a novel function for DBT. To avoid confusion, we agree with the reviewers’ concern about placing the S589D/4A PER isoform in our model. We have therefore removed the reference to S589D/4A in the schematic of our model (Figure 7).

We have also clarified the text to reflect our revisions: “This transition state [S589D/4A] can occur if S589 phosphorylation blocks PER-SD phosphorylation (Fu, 2008; Kivimäe et al., 2008) and may represent (i) a later stage isoform of PER at the end of transcriptional inhibition, (ii) an isoform that commits PER to degradation prematurely, or (iii) may not be an isoform relevant to the PER degradation pathway.”

B) It is likely that, in the absence of TIM, PER is rapidly degraded by DBT regardless of its phospho-state. We have included a western blot for the reviewers to support this claim (Author response image 1). We therefore have clarified our view regarding the presence of TIM and made the following changes:

a) We have added the sentence: “Thus, S589 phosphorylation has a stabilizing effect when TIM is present.” This emphasizes that TIM stabilization of PER cooperates with the S589 phospho-state.

b)We have added the sentence: “The changes in the stability of the PER variants correlate well with the differences in rhythmic behavior exhibited by the relevant fly mutants (Figure 1B).” This emphasizes the correlation of changes in behavior with the stability of each S589 PER variant.

c) We have removed the phrases “but that phosphorylation of S589 causes destabilization of PER in its absence” and “during the early stage of the circadian cycle (after CT2)”. We believe that this distracts from the emphasis we wished to place on S589 phosphorylation cooperating with TIM to stabilize PER.

C) With regard to why both S589D and S589A exhibit a short behavioral rhythm, the reviewers raise this issue in their review, section “a”, below. We offer a thorough explanation below. We have also modified the text to state, “The changes in the stability of the PER variants correlate well with the differences in rhythmic behavior exhibited by the relevant fly mutants (Figure 1B).” As mentioned above (B) to draw attention to the correlation between protein stability in neurons (Figure 2) and rhythmic behavior (Figure 1).

**Author response image 1. respfig1:** PER stability in steady state conditions in cultured cells. Wild type PER and PER variants S589A, S589D, 4A and 4D were exogenously expressed in cultured cells, or co-expressed with wild type DBT (**C**) or DBT-K38R (**K**). Protein extracts from cells were analyzed by immunoblot using antibody against myc tag (PER) or FLAG tag (DBT). The asterisk denotes a non-specific protein that cross reacts with the FLAG antibody. These data suggest that residue substitutions in the PERS and PER-SD regions do not block DBT-mediated PER degradation. TIM was not exogenously expressed in these experiments.

From Figure 6, S589D/4A is less stable then S589A/4A, so it might be possible that the S589D/4D becomes S589A/4A at some point, but a potential problem is that S589A/4A is too(?) stable-although it seems to be less stable then PER4A. Thus, the main problem I have is with the second part of the story; i.e., how does maintaining phosphorylation of 589 at the end of the cycle make it less stable then the wildtype, giving rise to a shorter period? In other words, they imply that de-phosphorylation of 589 late in the cycle is critical for delayed PER degradation. Why is that not in the model? And if 589 has to be dephosphorylated at some point, what is the PER species that is being degraded in the wildtype at the end of the cycle?

The reviewers here ask about the importance of S589 and how the phospho-state of S589 relates to the phosphorylation state of PER-SD in terms of PER stability. There is concern regarding the need for dephosphorylating S589 as a part of the mechanism of PER degradation.

We do not mean to imply that S589D/4D becomes S589A/4A, but instead suggest that S589D/4D may become S589D/4A or S589A/4D through de-phosphorylation. S589D/4A is a highly unstable isoform of PER (Figure 6C). The few S589D/4A variants that escape degradation and are available to revert back to the non-phosphorylated form (S589A/4A) would be fed through the same pathway until the number of PER molecules are reduced to levels insufficient to block CLK activity. S589A/4D on the other hand can easily be re-phosphorylated by DBT to form S589D/4D again. As we state in the manuscript, this detail of the proposed model is more speculative than other parts of the model that are directly supported by our data. Therefore, in order not to detract from the model, we removed S589D/4A from our schematic in Figure 7 (see also above). For the discerning reader who may wonder about this possible transition state, we have added a small section in the discussion on S589D/4A and clarified the text as follows:

“Phosphatases play a role in maintaining 24 hour behavioral rhythms (Chen et al., 2007; Fang et al., 2007). […] Thus the ordered phosphorylation of PER-SD and then S589 ensures that the circadian clock ticks forward.”

We also do not mean to imply that S589 is dephosphorylated as part of the mechanism of PER repression or PER degradation (Figure 7). We propose, based on data from Garbe et al. (2013) and Chiu et al. (2008), that the S589A/4D form is the predecessor isoform to the S589D/4D form. Indeed, S589 does not appear to be dephosphorylated as PER is being degraded, as shown by western blot by Chiu et al. (2011) from whole head protein extracts.

2) Also, the TIM switch argument is complicated. Basically, based on Figure 3, they say, "These data therefore suggest that S589 phosphorylation stabilizes PER in the presence of TIM from DBT-mediated degradation, either by blocking phosphorylation of other PER destabilizing sites, or in spite of phosphorylation of these other sites. In contrast, blocking S589 phosphorylation redirects DBT to promote the degradation of the PER/TIM complex, likely by phosphorylating other PER sites. Therefore, S589 acts as a switch, directing DBT to degrade PER or to stabilize PER within the PER/TIM complex."- While all of this is true, calling it a TIM switch is a bit confusing. All it means is what we already know, in the presence of TIM the 589D species is more stable than the 589A species when other sites are also phosphorylated. Otherwise stated, both 589A and 589D species are more stable with TIM, it is just that 589A is less stable then 589D. In do not see how 589A degradation is "TIM-independent", when it is clearly shown in Figure 3 that TIM stabilizes 589A (this was also shown by Li and Rosbash, 2013, JBR, and should be cited).

2a) The reviewers emphasize the confusion regarding the term “TIM switch”.

2b) The reviewers also raise the issue that S589A and S589D stabilization by TIM was previously known, and

2c) That it is unclear how S589A degradation could be “TIM-independent”.

2a) We recognize that there is some confusion regarding the term “switch”. Our data resolve a longstanding paradox of S589 mutation and its effect on period length by showing that S589 phosphorylation stabilizes and promotes PER inhibitory function in the presence of TIM and helps mediate PER degradation in the absence of TIM. We realize that the term “switch” does not capture this complex mechanism. Although we have not used the term “TIM switch”, we did call S589 a switch. To clarify this issue, we have removed most references to the term “switch” throughout the text and inserted more precise language. The text now reads:

“Our results indicate that phosphorylation of PER residue S589 stabilizes and activates PER inhibitory function in the presence of TIM, but promotes PER degradation in its absence.”

“The classic per short site S589 serves two functions: first to stabilize the activated PER inhibitor in the presence of TIM and then to mediate its degradation after TIM is degraded”

“Therefore S589 acts to direct DBT to degrade PER or to stabilize PER within the PER/TIM complex.”

“Thus, PER degradation through S47 can be regulated through two routes of phosphorylation, diverging with S589 phosphorylation.”

2b) It was not known before our study that S589D is more stable than the S589A variant, in the presence of TIM. To clarify the text and historical context, we have now included the citation suggested by the reviewers. The article cited by the reviewers (Li and Rosbash, 2013) use the original perS mutant, S589N, for their studies, not S589A. Substitution at this position would indeed block the ability of a kinase to phosphorylate the protein, but substitution with asparagine can lead to ambiguity. Asparagine has a dipole on its side chain. In this regard, it may be able to act as a weak phosphomimetic under some circumstances. It is for this reason we avoided including the S589N substitution in our studies, instead opting for the clearer S589A and S589D substitutions in PER.

Additionally, the paper by Li and Rosbash use PER/PER^S^ and TIM, without exogenous DBT expression in cultured cells. Since, in cultured cells, exogenous DBT is required for DBT phosphorylation of S589 (and other sites) (Chiu, Vanselow, Kramer, and Edery, 2008), we reasoned that endogenous DBT may not be sufficient to see all the effects of DBT-mediated degradation on PER and TIM that are driven by strong promoters in the Li and Rosbash study.

The text has been modified for clarification as follows:

We added the sentence “Analysis of the S589N variant of PER (PER^S^) in a similar study that included TIM (but not DBT) in cultured cells suggests that it is equivalent in stability to wild type PER (Li and Rosbash, 2013). However the side chain of asparagine does have a dipole that has the potential to act as a phosphomimetic while simultaneously blocking phosphorylation by DBT.”

2c) We did not mean to suggest that S589A degradation was “TIM-independent”. The source of this misunderstanding may have been: “In other words, these data suggest that while both S589A and S589D mutants exhibit shorter circadian periods because of their effects on PER stability (TIM independent and TIM dependent, respectively)” – The phrase in brackets has been removed for clarity. We instead meant that the stability of PER was variable with respect to wild type in the presence or absence of TIM. Figures 2D, 2E, 3A and B illustrate that the S589A PER variant and TIM are less stable compared to wild type PER/TIM in the presence of DBT. S589D on the other hand exhibits stability equivalent to wild type for both PER and TIM. Thus, we conclude that phosphorylation of S589 stabilizes the PER/TIM inhibitory complex. In the s-LNvs in vivo (Figures 2C-2E), the S589A variant of PER and TIM are less stable than wild type conditions after CT02, while the S589D variant demonstrates stability equivalent to wild type until TIM is degraded. When TIM is degraded, PER is rapidly turned over. In this regard, we conclude that S589 phosphorylation is stabilizing *before* TIM degradation, but becomes destabilizing *after* TIM degradation.

To remove any confusion about TIM-dependence with regard to the phospho-state of S589 or PER degradation, we have also removed or modified references to TIM dependence to place emphasis on S589 regulation of PER stability.

We removed the phrase in parentheses referring to TIM dependence, mentioned above.

We removed the words “TIM-dependent” in the abstract. The sentence now reads:

“Our results indicate that phosphorylation of PER residue S589 stabilizes and activates PER inhibitory function in the presence of TIM, but promotes PER degradation in its absence.”

To summarize:a) How does retaining S589 phosphorylation destabilize PER at the end of the cycle to get a shorter period compared to the wildtype situation as implied from the S589D mutant (i.e., at CT6 what is the wildtype form of PER that is more stable in comparison to flies that have 589D?). And if this implies that 589 has to be dephosphorylated at some point late in the cycle (so that it continues being stable and repressing for a few more hours), what is that species and why is it not shown in the model?

The reviewers ask for clarification on how S589D can be less stable than wild type PER at CT06, if wild type PER is phosphorylated to appear similar to S589D.

We do not know the precise phospho-state of PER at CT06 in the s-LNvs. Determining its form would require purification of PER from the s-LNVs and mass spectrometry analysis, which is beyond our current technical ability. We do not think that S589 is de-phosphorylated at the end of the PER life cycle. This also appears to be the case in Chiu et al., 2011, where they demonstrate that an S589 phospho antibody reacts with protein from fly head extracts to reveal S589 is phosphorylated even as PER is degraded.

However, it is clear from our data that S589D is less stable than wild type PER in the s-LNvs in the absence of TIM (CT06). This difference may lie in the variability of phosphorylation of this site in a biological system. Substitution of S589 with the phosphomimetic aspartate forces the protein to appear phosphorylated 100% of the time. In “normal circumstances”, it is possible for serine phosphorylation to often be incomplete, allowing for variation in any given process. We therefore suggest that the higher average stability of wild type PER at CT06 may be due to partial phosphorylation of sites that lead to PER degradation. The S589D substitution offers *certainty* of “phosphorylation”, accelerating the degradation process.

b) I do not fully understand the implication of calling the role of TIM vis. a vis. S589 a TIM switch when all phospho-isoforms of PER appear to be less stable without TIM in the presence of DBT. Indeed, from the model it appears that losing SD phosphorylation is required for degradation but not a change in the phosphorylation status of S589.

We hope we were able to address the issue raised about the term “switch” above.

We also hope that our discussion regarding S589D/4A and the changes we have made to the manuscript sufficiently addresses this question (described above). By removing the reference to S589D/4A isoform in our model, we place more emphasis on the S589D/4D variant of PER that suggests that DBT plays a role in PER function that has not been previously described and would appear contradictory to its known role in PER degradation: stabilizing the PER/TIM complex through phosphorylation and activation of PER inhibitory activity.

c) Chiu et al., 2011, showed that S589 is highly phosphorylated by ZT20, so it is not clear that there is an early species of S589A/4D, which is basically based on the Garbe et al., model. The Garbe model is based on epigenetics, so it is not a given that phosphorylation at SD blocks phosphorylation at S589. This potential discrepancy should be discussed. In any event, it was not discussed if there is a functional significance to the initial repressor being highly unstable, i.e., does it play a different role compared to the more stable 589D/4D?

We thank the reviewer for this insightful comment. Yes, the evidence in Chiu et al. imply that the S589A/4D transition state may be a short-lived isoform. While we included S589A/4D in the isoforms tested because of its pivotal role in the Garbe model, our data do not support a different role than the more stable 589D/4D, which we did not address explicitly in the initial manuscript. In response to the reviewer, we have amended the discussion to include a brief discussion of this issue. The Discussion has been modified as follows:

“In vivo, S589 is phosphorylated at ~ZT20, and appears to remain phosphorylated as PER is degraded (Chiu et al., 2011). […] There is no apparent need for an unstable PER protein early in transcriptional inhibition and it is more likely that both the S589 and PER-SD sites cooperate to regulate PER stability and function.”

Therefore, while the role of 589D/4D as a novel, stable and strong repressor is well documented and highly interesting, and an added role for 589 phosphorylation status in the degradation of PER after CT2 is also interesting, the latter part of this story and a specific role for TIM are less solid, making the latter part of the model unclear as to how it was derived based on the data.When presenting the model in Figure 7, the authors need to present their arguments for the time course and nature of the phosphorylated species at each step, with alternatives presented if possible.

We are grateful that the reviewers appreciate our novel contributions on the role of S589 phosphorylation state in regulating PER function and stability and a new role for DBT in stabilizing the PER/TIM inhibitory complex. The reviewers point out that a specific role for TIM in our model is less clear.

We have altered our model based on the suggestions made by the reviewers (see above and also below). We have further clarified the text with alternatives presented, particularly for the role of the 589D/4A isoform (see above). We hope that the changes made satisfy the concerns raised.

We have altered our model based on the suggestions made by the reviewers and amended the Discussion: “DBT begins phosphorylation of PER in the PER-SD (Garbe et al., 2013; Kivimäe et al., 2008) (Step 1), initiating inhibition of dCLK-mediated transcription by PER. […] Thus, the steps that delay dCLK-mediated transcription activity rely on the activation and stabilization of PER by DBT phosphorylation, before TIM degradation ensues.”

We have changed the figure legend to reflect the modified model: “Non-phosphorylated PER is stable, but unable to repress CLK-mediated transcription. […] After CT06, TIM is degraded, releasing PER for degradation, ending the PER life cycle.”

3) The authors largely ignore related work in mammalian systems showing that CKId has different effects on Per2 depending on the site of phosphorylation. The PER phosphorylation cascade downstream of the classic per short site likens to the phosphorylation cascade reported in mammalian Pers (Xu et al., 2007). This paper should be cited and discussion regarding similarities and differences of the reported findings with these published data should be addressed.

The reviewers suggest citation and discussion of the work by Xu et al. This is an excellent point and we apologize for this oversight.

We have now amended the Discussion to highlight this paper and address the similarities and differences between the *Drosophila* and mammalian regulation of PER via phosphorylation. In summary, while there appear to be differences in specific effects (the phospho-state of mammalian PER S662 regulates repression activity rather than protein stability), both mammals and flies appear to share in common the principle of distinct mechanisms by which DBT/CK1d regulates activity and stability of PER. The text has been modified to include the following paragraph in the Discussion, centered on this very interesting issue:

“Distinct phospho-programs also regulate human PER2 stability and inhibitor activity (Xu et al., 2007). […] Thus, despite differences in the PER function that sites S662 and S589 regulate, both mammals and flies utilize distinct mechanisms that employ DBT/CKIδ to separately regulate the activity and stability of Period protein, delaying transcription reactivation in the circadian clock.”

4) Another interesting observation are that the results with the S589A and S589D alleles don't all agree with what might be predicted regarding expected DBT phosphorylation effects (assuming 589D mimics a constitutively phosphorylated S589). It would be worth also looking at 2 additional papers: (Kaasik et al., 2013; Li et al., Cell Metabolism 2013). These papers show that the same Serines in the "cascade region" in mammals are also being O-GlcNAC modified. That is, CKId and OGT are competing for occupancy and modification of the same sites. An interesting question is whether O-GlcNAC modified vs. phosphorylated Serines in this region have different downstream effects. Has anyone looked to see whether the phosphorylation cascade region in dper is O-GlcNAC modified? While the fly and mammalian regions are not really conserved at a sequence level, the similarity of some of the effects in fly suggests they may be similar functionally. If so, then phosphorylated vs. GlcNAC modified vs. unmodified may underlie the unexpected results of S589 vs. D589.

The reviewers bring up the possibility that the S589 site is GlcNAc modified, as an explanation of why S589A and S589D do not exhibit opposing changes in behavioral rhythmicity.

We agree that it is surprising that the phosphomimetic mutant S589D does not give the inverse phenotype of the S589A mutant, but we hope that we were able to make the case for S589D. S589D, while stabilized in the presence of TIM, is degraded more rapidly (relative to wild type) in the absence of TIM.

We considered the possibility of GlcNAc modification of this region. We have successfully used the S589A and S589D PER mutants to block the phenotypic effects of DBT overexpression in flies (Figure 1C), demonstrating that there is a genetic interaction between S589 and DBT, and modification of S589 is rate limiting. We conducted a similar experiment in which we overexpressed O-GlcNAc Transferase (OGT) in the different S589 mutant backgrounds. In this case, we were not able to demonstrate any genetic interaction between S589 and OGT. We therefore concluded that O-GlcNAc-ylation was unlikely to be involved in regulating the function of S589.

We have included these data as supplemental data in the Results section to address this concern by the reviewers:

“The activity of *O*-GlcNAc transferase (OGT) has been shown to influence behavioral rhythmicity in mammals and flies, either by competing with CKIδ for modification of mPER2 (orthologues of DBT and PER, respectively) or blocking BMAL1/CLOCK (orthologues of CYC/dCLK in flies) ubiquitination in mice, or prolonging PER stability in flies (Kaasik et al., 2013; Kim et al., 2012; Li et al., 2013). […] The same experiment conducted in the S589A and S589D mutant backgrounds similarly lengthened rhythmic behavior of the mutant flies (Figure 1—figure supplement 1). We therefore conclude that S589 does not genetically interact with OGT.”

We have also included additional information in the Materials and methods section: “UAS-OGT transgenic flies were a gift from Louis Ptáček (University of California, San Francisco). Flies overexpressing OGT were generated by recombining the second chromosome carrying the tim-UAS-Gal4 and UAS-OGT transgenes.”

5) There is no need for the SAXS data, etc. as it does not add anything to the Discussion.

While we agree that the SAXS data do not add positive results to our model, these and other biophysical data were included to rule out the possibility that the mutations studied might alter multimerization or cause other global conformational changes, such as gross disruption of protein folding. Our data suggest that they may instead alter local or other subtle conformational changes. If the reviewers agree, we would like to keep this information in supplemental to address such concerns that other readers may have regarding the overall effects of our residue substitutions.